# Poly(fluorenyl aryl piperidinium) membranes and ionomers for anion exchange membrane fuel cells

Nanjun Chen [1,5], Ho Hyun Wang[1,5], Sun Pyo Kim[1], Hae Min Kim[1], Won Hee Lee[1], Chuan Hu[1], Joon Yong Bae[1], Eun Seob Sim[2], Yong-Chae Chung[2], Jue-Hyuk Jang[3], Sung Jong Yoo [3], Yongbing Zhuang [4] & Young Moo Lee [1 ✉]

Low-cost anion exchange membrane fuel cells have been investigated as a promising alternative to proton exchange membrane fuel cells for the last decade. The major barriers to the viability of anion exchange membrane fuel cells are their unsatisfactory key components— anion exchange ionomers and membranes. Here, we present a series of durable poly(fluorenyl aryl piperidinium) ionomers and membranes where the membranes possess high $OH^-$ conductivity of 208 mS $cm^{-1}$ at 80 °C, low $H_2$ permeability, excellent mechanical properties (84.5 MPa TS), and 2000 h ex-situ durability in 1 M NaOH at 80 °C, while the ionomers have high water vapor permeability and low phenyl adsorption. Based on our rational design of poly (fluorenyl aryl piperidinium) membranes and ionomers, we demonstrate alkaline fuel cell performances of 2.34 W $cm^{-2}$ in $H_2$-$O_2$ and 1.25 W $cm^{-2}$ in $H_2$-air ($CO_2$-free) at 80 °C. The present cells can be operated stably under a 0.2 A $cm^{-2}$ current density for ~200 h.

[1] Department of Energy Engineering, College of Engineering, Hanyang University, Seoul, Republic of Korea. [2] Department of Materials Science and Engineering, College of Engineering, Hanyang University, Seoul, Republic of Korea. [3] Hydrogen Fuel Cell Research Center, Korea Institute of Science and Technology (KIST), Seoul, Republic of Korea. [4] State Key Laboratory of Biochemical Engineering, Institute of Process Engineering, University of Chinese Academy of Sciences, Chinese Academy of Sciences, Beijing, PR China. [5] These authors contributed equally:Nanjun Chen,Ho Hyun Wang. ✉email: ymlee@hanyang.ac.kr

Anion exchange membrane fuel cells (AEMFCs) have made significant advances, especially in power density, that represents a parallel track to proton exchange membrane fuel cells (PEMFCs)[1–3]. Operating in alkaline environments, AEMFCs possess evident cost advantages due to the possibility of using non-platinum group metal (non-PGM) catalysts. AEMFCs intend to address high-cost issues associated with PEMFCs and to raise the economic competitiveness of low-temperature fuel cells with other power generation technologies in many burgeoning areas, particularly in light-duty transportation[4–8]. However, the major barriers to the viability of AEMFCs are their key but unsatisfactory materials including anion exchange polyelectrolytes (AEPs), which can be used both as anion exchange ionomers (AEIs) and membranes (AEMs)[9–12]. AEPs consist of polymer backbones and the pendent cationic groups that act as ion transporting species to conduct $OH^-$ and to transport water molecules[13–16]. Although numerous cationic groups (ammonium[17], imidazolium[15,16], phosphonium[18], sulfonium[19], and organometallic cation[20,21]) and polymer backbones[22–28] have been employed in AEPs so far, only a few AEPs display satisfactory performance at high pH and temperature (>80 °C)[17,29–33].

To date, years of study have revealed that the ammonium group has been the most-studied cationic group for AEPs due to its high ion conductivity, durability, and practicability. Moreover, the discovery of aryl ether-free AEPs addressed several early-stage problems of AEMs, such as low ion conductivity, insufficient alkaline stability, and poor mechanical properties[17,29,32]. Three types of aryl ether-free AEPs are dominant in AEMFCs so far: benzyl trimethylammonium (BTMA)[12,29,30], alkyl ammonium[10,34,35], and dimethylpiperidinium (DMP)-type AEPs[17,32,36]. Wang et al.[29] reported an AEMFC based on BTMA-type poly(ethylene-co-tetrafluoroethylene) (BTMA-ETFE) ionomers and high-density polyethylene (BTMA-HDPE) AEMs that reached a peak power density (PPD) over 2 W cm$^{-2}$ at 80 °C in H$_2$–O$_2$ with a 0.70 mg cm$^{-2}$ Pt–Ru/C anode. Mandal et al.[31,33] reported that H$_2$–O$_2$ AEMFCs based on BTMA-ETFE ionomers and PTFE-reinforced poly(norbornene) (PNB) AEMs reached a PPD over 3 W cm$^{-2}$ at 80 °C with a 0.70 mg cm$^{-2}$ Pt–Ru/C anode. However, BTMA-ETFE possessed poor alkaline stability because the BTMA-ETFE membrane was brittle after being soaked in 1 M NaOH at 80 °C for only 168 h[12]. Lee et al.[35] and Maurya et al.[34] reported alkyl ammonium poly(terphenylene) AEMs and alkyl ammonium fluorene ionomers.

Alkyl ammonium-type AEPs[34,37] exhibited preferable alkaline stability (stable in 1 M NaOH at 80 °C for 720 h) compared to BTMA-type AEPs, while the PPD of these AEMFCs was limited to below 1.6 W cm$^{-2}$. Wang et al.[17] also reported stable poly(biphenyl piperidinium) (PFBP-0) ionomers and copoly(aryl piperidinium) (c-PAP) AEMs. Their H$_2$–O$_2$ AEMFCs reached a PPD of 1.89 W cm$^{-2}$ with a 0.7 mg cm$^{-2}$ Pt–Ru/C anode. However, biphenyl or terphenyl groups in PAP AEIs have been documented to possess high phenyl adsorption on catalysts[34,37–39]. Currently, AEI research has been highlighted as a crucial issue by the latest US Department of Energy (DOE) protocol for the next decade. However, we lack sufficient insight from ionomer research, which has not revealed the effects of water vapor transport behavior and molecular dimensions of AEIs on the cell performance.

Here, we present poly(fluorenyl aryl piperidinium) (PFAP) copolymers for AEMs and AEIs. The fluorene (FLN) segment in the copolymers improves the rigidity and phase-separated morphology of AEMs to increase the dimensional stability and ion conductivity. Moreover, AEIs with rigid FLN groups are expected to improve the water vapor permeability (or water diffusivity) and decrease the phenyl adsorption effect.

## Results

**Design of polymers and characterization.** Our PFAP copolymers were synthesized by a facile acid-catalyzed condensation reaction (Supplementary Fig. 1). Poly(fluorene-co-biphenyl N,N′-dimethylpiperidinium) (PFBP-$x$) and poly(fluorene N,N-dimethylpiperidinium-co-nonafluoride) (PFPN-$x$) are designed for AEIs (Fig. 1a, b), whereas poly(fluorene-co-terphenyl N,N′-dimethylpiperidinium) (PFTP-$x$) is specifically considered for AEMs (Fig. 1c). Here, $x$ is the molar ratio of the fluorenyl piperidinium segment in the copolymer. Structural analysis of PFAP-$x$ can be confirmed by [1]H nuclear magnetic resonance ([1]H NMR) spectra (Supplementary Figs. 2–12).

PFAP AEIs were rationally designed to combine the merits of currently representative BTMA-ETFE[29,30], alkyl ammonium PF[34], and PAP ionomers[17,36], and overcome their drawbacks. They have advantages including (1) PFBP AEIs containing DMP and FLN groups possess high ex-situ durability, rigidity, and low ionomer phenyl adsorption. Torsional rotation calculations demonstrated that FLNs had much smaller dihedral angle in the optimized

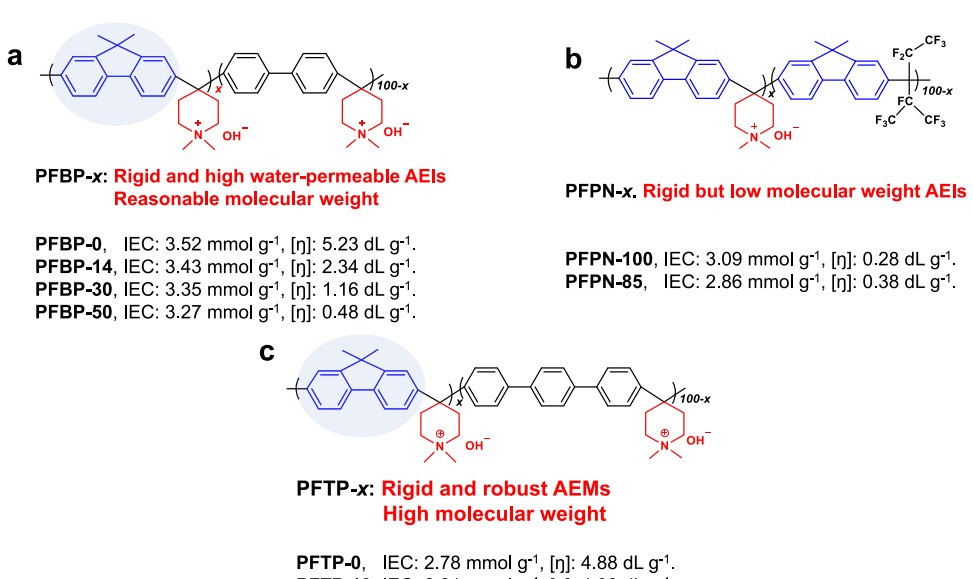

**PFBP-$x$: Rigid and high water-permeable AEIs**
**Reasonable molecular weight**

PFBP-0,  IEC: 3.52 mmol g$^{-1}$, [η]: 5.23 dL g$^{-1}$.
PFBP-14, IEC: 3.43 mmol g$^{-1}$, [η]: 2.34 dL g$^{-1}$.
PFBP-30, IEC: 3.35 mmol g$^{-1}$, [η]: 1.16 dL g$^{-1}$.
PFBP-50, IEC: 3.27 mmol g$^{-1}$, [η]: 0.48 dL g$^{-1}$.

**PFPN-$x$. Rigid but low molecular weight AEIs**

PFPN-100, IEC: 3.09 mmol g$^{-1}$, [η]: 0.28 dL g$^{-1}$.
PFPN-85,  IEC: 2.86 mmol g$^{-1}$, [η]: 0.38 dL g$^{-1}$.

**PFTP-$x$: Rigid and robust AEMs**
**High molecular weight**

PFTP-0,  IEC: 2.78 mmol g$^{-1}$, [η]: 4.88 dL g$^{-1}$.
PFTP-13, IEC: 2.81 mmol g$^{-1}$, [η]: 4.08 dL g$^{-1}$.

**Fig. 1 The chemical structure of representative polyelectrolytes with IEC and intrinsic viscosity. a** PFBP-$x$, **b** PFPN-$x$, and **c** PFTP-$x$.

geometry along with higher rotation energy barrier than biphenyl and terphenyl groups (Supplementary Fig. 13), implying the high rigidity of FLNs[40]. Density functional theory (DFT) calculations revealed that FLN-based molecules have lower phenyl adsorption energies than those of biphenyl-based molecules on Pt or Pt–Ru (111) crystal planes (Supplementary Fig. 14). (2) PFAP AEIs possess better solubility than ordinary PFBP-0 and PFTP-0 ionomers due to the presence of rigid FLN groups. Solubility testing of these polymers demonstrated that PFBP ionomers showed the best solubility in IPA/DI water (Supplementary Table 1), which is beneficial for catalyst slurry preparation.

As for the advantages of PFAP AEMs, (1) FLNs increase the rigidity of PFTP AEMs, which simultaneously improves the ion-conducting capability and dimensional stability (Supplementary Fig. 15). PFTP-13 AEMs with high IEC show lower water SRs than those of PFTP-0 AEMs (Supplementary Table 2). (2) PFAP copolymers have facile synthetic processes (two steps, no heating, short synthetic period). PFAP AEIs and AEMs possess significantly different intrinsic viscosities ($[\eta]$) depending on the FLN content (Supplementary Table 3), which is related to the differences in their molecular weights. PFBP-14 (2.32 dL g$^{-1}$), PFTP-13 (4.08 dL g$^{-1}$), PFBP-0 (5.25 dL g$^{-1}$), and PFTP-0 (4.875 dL g$^{-1}$) possess a high $[\eta]$ that provides excellent film-forming properties. PFAP-$x$ and PFPN-$x$ copolymers with high FLN contents exhibit low $[\eta]$ resulting in limited film-forming properties, while the $[\eta]$ of PFPN (~0.38 dL g$^{-1}$) is still close to Olsson et al.'s PAPs[24] (~0.2 to ~0.47 dL g$^{-1}$). PFAP copolymers with high $[\eta]$ exhibit preferable film-forming properties.

**Water sorption and transport behavior and gas permeability.**
Water management plays a crucial role in AEMFCs. Water is a reactant in the cathode and a product in the anode, and the amount of water generated in the anode is two times faster than the electrochemical consumption of water in the cathode (Fig. 2a). This is why it is easy to flood the anode, while the cathode is inclined to dry out. Different PFAP-$x$ polymers exhibit significantly different water uptakes (WUs) and swelling ratios (SRs) with different counterions (Fig. 2b, Supplementary Fig. 16a, b). PFTP-13 (WU: ~45%, SR: ~16%) and PFTP-0 (WU: ~55%, SR: ~24%) membranes exhibited much lower WU and SR in liquid water at 30 °C compared to PFBP-14 (WU: ~300%, SR: ~100%) and PFBP-0 (WU: ~350%, SR: ~110%) due to lower IEC values. Meanwhile, the PFTP-13 membrane with higher IEC displayed lower SR and enhanced dimensional stability compared to the PFTP-0 membrane due to the presence of rigid FLN groups.

Dynamic water vapor sorption (DVS) of PFAP-$x$ membranes is significantly lower than their liquid WU (in equilibrium) at different RHs (Fig. 2c). PFBP-14 and PFTP-13 films exhibited lower water vapor sorption than PFBP-0 and PFTP-0 films, respectively, which is consistent with liquid WU behavior. Notably, WU represents the water sorption capacity of AEIs or AEMs when liquid water is forming in AEMFCs, while it cannot represent the water transport behavior. Therefore, water diffusivity of AEIs at different RHs was automatically estimated by DVS, as shown in Supplementary Table 4. PFBP-14 and PFBP-0 AEMs with higher IEC values exhibited slightly higher water diffusivities than those of PFTP-13 and PFTP-0 AEMs. PFBP-14 and PFTP-13 AEMs also display higher water diffusivities than those of PFBP-0 and PFTP-0 AEMs, respectively, due to the existence of FLN blocks.

Currently, the water transport behavior can be referred to as water diffusion or water permeability related to AEMs[41,42]. Three small gas molecules—H$_2$, O$_2$, and water vapor—reach and participate in catalytic reactions, yet none of the current reports has accurately analyzed the water vapor permeability ($P_{water}$)

through AEMs or AEIs at different RHs. Similar to water diffusivity behavior, water vapor permeabilities of PFBP-14 and PFBP-0 are 62,398 Barrer (where 1 Barrer = $10^{-10}$ cm$^3$ (STP) cm cm$^{-2}$ s$^{-1}$ cm Hg$^{-1}$) and 58,921 Barrer at 85% RH, respectively, and those of PFTP-13 and PFTP-0 are 20,000–25,000 Barrer (Fig. 2d). Note that BP-containing copolymers (PFBP-0 and PFBP-14) showed high water vapor sorption and much higher $P_{water}$ compared to TP containing AEPs (PFTP-13 and PFTP-0). FLN-containing PFAP copolymers (PFBP-14 and PFTP-13) with lower water vapor sorption and liquid WU exhibited similar or even higher water vapor permeability than that of PAP AEPs (PFBP-0 and PFTP-0) due to the rigid FLN groups, the unique phase-separated morphology, and large water channels (Supplementary Fig. 17). The detailed discussion of membrane morphology and water channels are presented in Supplementary information.

On the other hand, the H$_2$ and O$_2$ permeabilities of AEPs were measured and recorded at different RHs using Barrer as a unit that is well-known in gas separation communities[43] (Fig. 2e and Supplementary Fig. 16c, d). PFTP-13 and PFTP-0 exhibited much lower H$_2$ (~10 Barrer) and O$_2$ (<0.5 Barrer) permeabilities compared to PAP-TP-85 (H$_2$: ~35 Barrer)[17], commercial FAA-3-50 (H$_2$: ~15 Barrer), indicating that the PFTP-13 and PFTP-0 AEMs possess superior gas barrier properties.

**Electrochemical and physical properties.** FLN-based PFAP copolymers exhibited superior ion conductivity compared to PAP AEPs (Fig. 3a, b). Among these AEPs, PFTP-13 AEPs with the lowest swelling ratio displayed the highest ion conductivities of 208 and 77 mS cm$^{-1}$ in OH$^-$ and Cl$^-$ forms, respectively, at 98 °C. An appropriate FLN segment incorporated into the PFAP backbone is beneficial for improving ion conductivity. Supplementary Fig. 17 shows that PFTP-13 and PFBP-14 AEPs possess preferable microphase separated morphologies and larger water channels compared to PFTP-0 and PFBP-0 AEPs, which improves their ion conductivity and water vapor permeability. The CO$_3^{2-}$ conductivities of PFTP-13 and PFBP-14 were over 65 mS cm$^{-1}$ at 80 °C (Supplementary Fig. 18), meaning that the PFTP-13 and PFBP-14 still possess high ion conductivity after carbonation[13]. Differential scanning calorimetry (DSC) analysis showed that excessive free water content ($N_{free}$) in PAP AEPs did not enhance the ion conductivity (Supplementary Fig. 19 and Table 2), indicating that the ion conductivity of swollen PAP AEMs does not always match well with IEC, but is related to the morphology and water transport behavior[17,31].

PFTP-13 exhibits high tensile strength (TS, 84.6 MPa), elongation at break (EB, 25.6%), and Young's modulus (YM, 1580 MPa) among these AEPs, and it can be easily fabricated into thin membranes (7–20 μm) (Fig. 3c–g and Supplementary Fig. 20). Supplementary Table 5 lists the mechanical properties of AEMs reported to date. The mechanical properties of the PFTP-13 AEMs without reinforcement are close to the best in current research. PFAP-13 AEMs possess high glass-transition temperatures ($T_g$) and an excellent storage modulus (SM) ($E'$) over 1500 MPa at 80 °C (Supplementary Figs. 21 and 22) that is the highest value so far in AEMs. This implies that PFTP-13 AEMs possess high rigidity and thermomechanical stability. Note that PFAP copolymers show two $T_g$s, which are due to the two different segments in the polymer backbone. In contrast, homopolymers such as PFTP-0 and PFBP-0 exhibit only one $T_g$. On these grounds, the PFTP-13 copolymer with low SR, excellent dimensional stability, and high ion conductivity was chosen as representative AEMs, while PFBP-14 with high ion conductivity, high water permeability (or water diffusivity), but limited dimensional stability was selected as representative AEIs in this work.

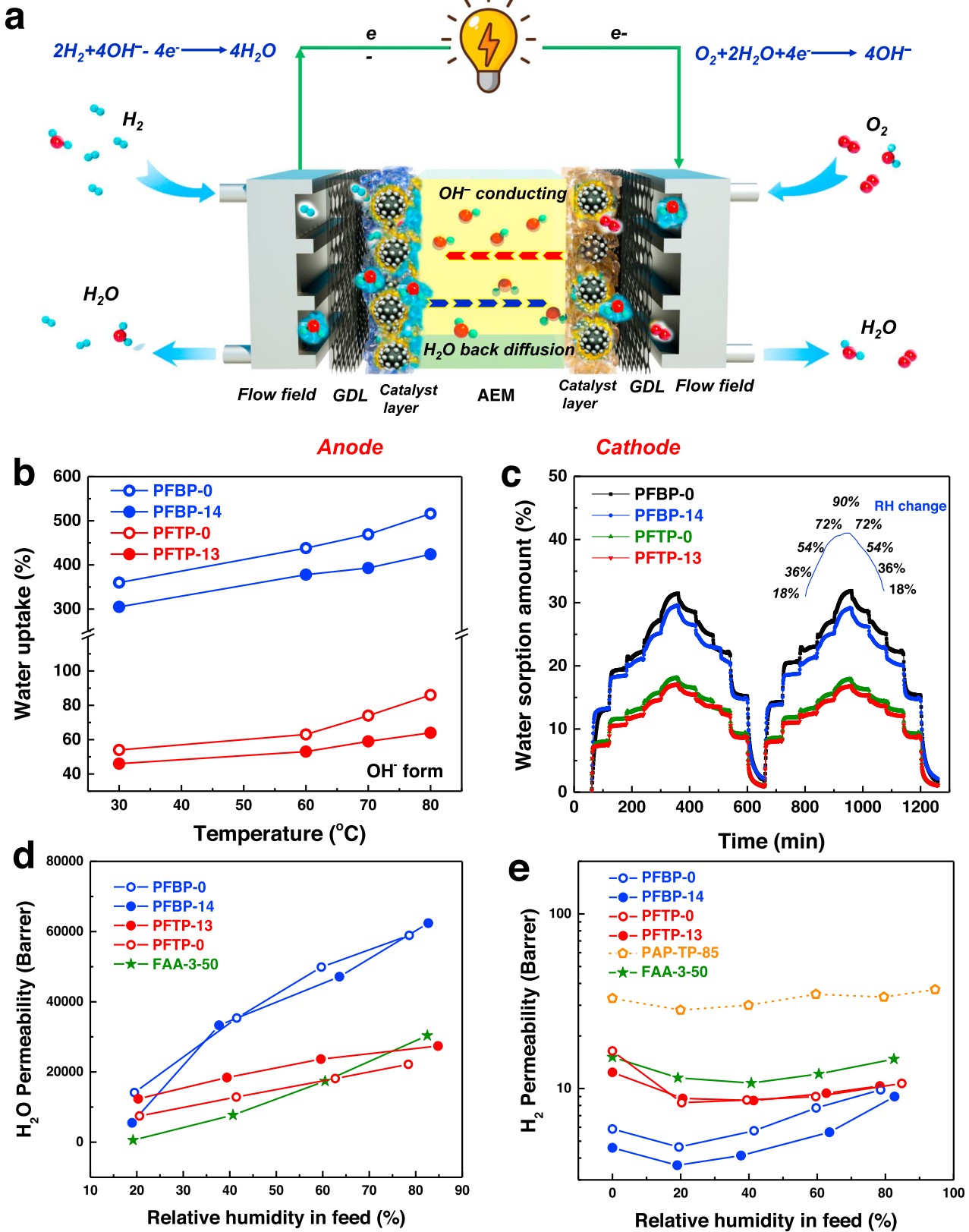

**Fig. 2 Gas and water sorption and transport behavior. a** Schematic diagram of the membrane electrode assembly (MEA) in AEMFCs emphasizing fuel gas and water transport. **b** The WU of PFBP-0, PFBP-14, PFTP-0, and PFTP-13 membranes in OH⁻ form in liquid water. **c** Dynamic water vapor sorption behavior of PFBP-0, PFBP-14, PFTP-0, and PFTP-13 membranes at different RHs measured in a DVS instrument (RH changes between 18%, 36%, 54%, 72%, 90% at a time interval of 60 min) at 25 °C. RH is controlled to automatically increase in a DVS instrument at a given time interval. Two hydration–dehydration cycles were recorded. **d** Water vapor permeability, and **e** H₂ permeability of different AEPs in I⁻ form at 60 °C under 2.2 bar unilateral backpressure. H₂ permeability of reported PAP-TP-x[17] is presented for comparison.

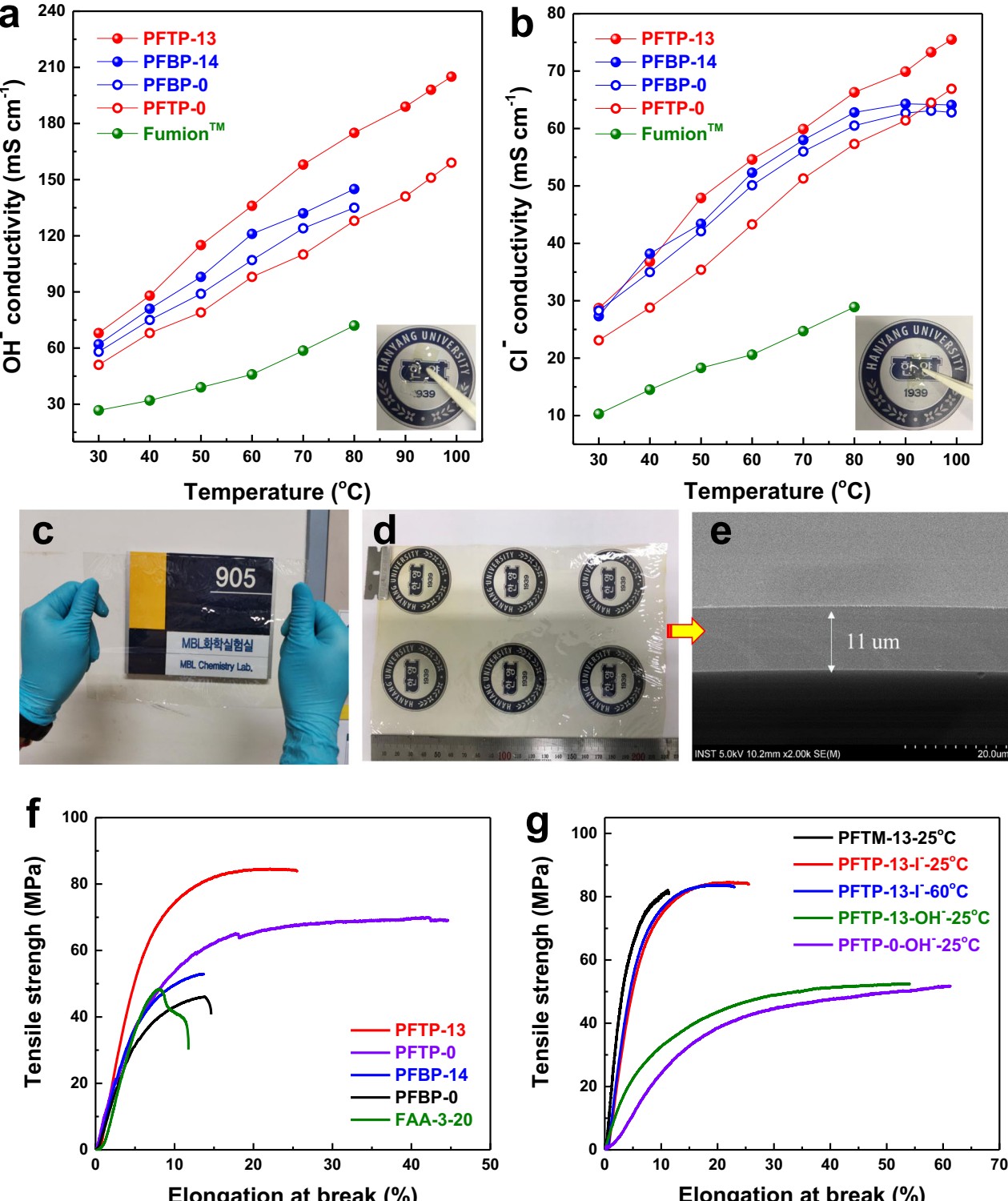

**Fig. 3 Physical properties of AEMs. a** OH⁻ and **b** Cl⁻ conductivity of different AEMs as a function of temperature along with the picture of PFTP-13 membranes after testing at 98 °C. PFBP-14 and PFBP-0 membranes showed poor dimensional stability at high temperatures, and thus it was difficult to measure their OH⁻ conductivities over 90 °C. **c**, **d** A picture of transparent PFTP-13 membranes 13.5 × 22.5 cm in size. **e** SEM image of a cross-section of PFTP-13 membrane with a thickness of ~15 μm. **f** The TS and EB of AEMs in I⁻ form and commercial FAA-3-20 membranes at room temperature. Compared to the present PFTP-0 AEMs (TS: 71 MPa, EB: 45.7%, YM: 1.2 GPa) and reported PAP-TP-x AEMs (TS: 67 MPa and EB:117%)[17], PFTP-13 AEMs exhibit much higher TS and YM but lower EB, indicating that the PFTP-13 AEMs have a higher deformation resistance. **g** The tensile strength and elongation at break of PFTP-13 and PFTP-0 membranes at different temperatures and types. PFTM-13 is PFTP-14 before quaternization, whereas PFTP-13-I⁻ and PFTP-13-OH⁻ are in I⁻ and OH⁻ forms, respectively. PFTP-13 AEMs maintain their mechanical properties at 60 °C. Compared to Peng et al.'s PFTP-0 AEMs[36] (TS: 35 MPa and EB: 40%), the present PFTP-13 and PFTP-0 AEMs (TS: > 50 MPa and EB: ~60%) in OH⁻ form exhibit much higher mechanical properties due to their higher intrinsic viscosity.

**AEMFC performance**. The effect of water behavior of AEIs on AEMFC performance was systematically investigated at different RHs based on the commercial FAA-3-20 and PFTP-13 AEMs with TKK Pt/C. At high anode/cathode (A/C) RHs (90%/100%) and low current density (<2.5 A cm$^{-2}$), Supplementary Fig. 23a showed that PFTP-13 (WU: ~50%) and PFBP-14 (WU: ~300%) ionomers exhibited similar PPDs at 65 °C based on the commercial FAA-3-20 membrane, while those of FLN-based ionomers show higher PPDs than PFBP-0 (WU: ~350%) and PFTP-0 (WU: ~55%) ionomers due to the lower phenyl adsorption (Supplementary Fig. 23b) and higher ion conductivity. On the other hand, at high RH and moderate current density (~3.5 A cm$^{-2}$) (Fig. 4a) or at low RHs (50%/80%) and high current density (>5 A cm$^{-2}$) (Fig. 4b), PFBP-14 (1.42 W cm$^{-2}$) and PFBP-0 (1.19 W cm$^{-2}$) exhibited higher PPDs than PFTP-13 (1.06 W cm$^{-2}$) and PFTP-0 (0.98 W cm$^{-2}$) ionomers due to lower phenyl adsorption and higher $P_{water}$ (or high water diffusivity). According to Matanovic et al.'s discovery[37,38], poly(biphenyl)-based ionomers showed higher PPDs than poly (terphenyl)-based ionomers due to lower phenyl adsorption, which is basically matching with our results. Moreover, high $P_{water}$ of PFBP-14 AEIs with moderate WU contributed to rapidly discharging the generated water in the anode which improved the water back diffusion or to maintaining the water content at low RH conditions. PFBP-14 ionomers (~300% WU at room temperature) showed much lower WU compared to the state-of-the-art PF ionomers[34] (side-chain-type FLN-100, WU > 2000% at room temperature) at similar IECs (~3.45 mmol g$^{-1}$). Therefore, PFBP-14 ionomers with moderate WU did not raise the anode flooding issues in the present AEMFCs. On the contrary, PFTP-13 ionomers with low WU and high conductivity showed limited AEMFC performance at low RH due to relatively low $P_{water}$. Meanwhile, AEMs also require reasonable water permeability. Note that PFTP-13 AEMs display higher PPDs than the commercial FAA series[14] and PFTP-0 AEMs due to higher ion conductivity, mechanical properties, and $P_{water}$ (or water diffusivity) (Supplementary Fig. 24).

Unfortunately, PFPN-100 and PFPN-85 AEIs with low phenyl adsorption and high rigidity (Supplementary Figs. 13 and 14) possess insufficient intrinsic viscosity (or molecular weight), which cannot tightly hold on catalyst particles, resulting in severe catalyst detachment from catalyst-coated membrane (CCM) during ion-exchange process. Therefore, AEMFC performance of PFPN-$x$ ionomers (0.3–0.7 W cm$^{-2}$) is limited. Figure 4c shows that the PPDs of AEMFCs are significantly impacted by the [η] of PFBP-$x$ AEIs, and the detachment issue has been found when the [η] is lower than 1 dL g$^{-1}$ along with PPD decrease. Development of PFPN-$x$ and PFBP-$x$ ionomers with reasonable molecular weight will be our future work. These discoveries provide a clear information for the PAP family to rationally design AEIs with sufficient molecular weight. PFBP-$x$ AEIs containing 14–30% FLNs displayed the highest PPDs (1.52–1.64 W cm$^{-2}$) due to the reasonable [η] and high $P_{water}$.

Based on PFBP-14 AEIs, PFTP-13 AEMs, and 75%/100% A/C RH, the PPDs of AEMFCs reached 1.67 W cm$^{-2}$ (0/0 bar back pressure) and 2.34 W cm$^{-2}$ (1.3/1.3 bar back pressure) at 80 °C based on Pt–Ru/C anode (Fig. 4d). A small applied back pressure (0.5/0.5 bar) shows significant effect on the cell performance due to the higher electrode reactions and optimizing water transport that decreases the mass transport resistance (Fig. 4d and Supplementary Fig. 24b). Under H$_2$–air (CO$_2$ free) conditions, the present AEMFCs reached PPDs of 1.25 and 1.01 W cm$^{-2}$ with 0.42 mg cm$^{-2}$ Pt–Ru/C anode and 0.33 mg cm$^{-2}$ Pt/C anode, respectively (Fig. 4e). Moreover, after replacing the expensive Pt/C cathode with a home-made cobalt catalyst (0.6 mg cm$^{-2}$ Co@C/C, 40 wt% Co), the PFAP-based AEMFCs still reached an impressive PPD of 0.891 W cm$^{-2}$ at 80 °C with a low back pressure (Fig. 4f).

Optimized AEMFCs based on the Co@C/C cathode will be presented in our following paper.

Supplementary Table 6 summarizes all the fuel cell performances with low catalyst loadings conducted in this work, and these are compared with those in the literature. Currently, the state-of-the-art PTFE-reinforced PNB (GT-$x$ series: 3.2 W cm$^{-2}$ in H$_2$–O$_2$ and 1.75 W cm$^{-2}$ in H$_2$–air without back pressure[2]; 3.5 W cm$^{-2}$ in H$_2$–O$_2$ and 1.25 W cm$^{-2}$ in H$_2$–air with 0.5/1.0 bar back pressure[31]) and BTMA-HDPE (2.35–2.5 W cm$^{-2}$ in H$_2$–O$_2$ and 1.06 W cm$^{-2}$ in H$_2$–air without back pressure)[3,29] AEMs lead the current AEMFCs, while our present AEMFCs display comparable power density (2.34 W cm$^{-2}$ in H$_2$–O$_2$ and 1.25 W cm$^{-2}$ in H$_2$–air with 1.3/1.3 bar back pressure). On the other hand, compared to the state-of-the-art polyaromatics-based AEMFCs (<2.08 W cm$^{-2}$ in H$_2$–O$_2$ with 2.0/2.0 bar back pressure)[17,32,34], our present AEMFCs showed higher PPDs. Note that GT-$x$ series and BTMA-HDPE-based AEMFCs employed a high Pt–Ru/C loading of 0.7 mg cm$^{-2}$, while the present PFAP-based cells exhibited high performance with a low PGM loading of 0.33–0.42 mg cm$^{-2}$ [44,45].

Figure 5 summarizes the PPD and OH$^-$ conductivities of representative AEIs in current research. Compared to side-chain-type PF AEIs and poly(terphenylene) AEMs, PFBP-14 AEIs and PFTP-13 AEMs showed higher ion conductivity and PPDs. Although BTMA-ETFE ionomers displayed the slightly higher PPDs, the alkaline stability issues associated with the BTMA-ETFE still have not been well addressed. Our present ion conductivity and PPDs are the topmost values to date, compared to polyfluorene (PF), polyphenylene (PP) and PAP AEIs. This indicates that PFAP AEPs are good candidates for AEI and AEM applications.

**Ex-situ and in-situ durability**. Based on the high-throughput testing of current AEMs by the National Renewable Energy Laboratory (NREL)[8], most AEMs show severe deformation of the membranes and even exhibit fracture along with color changes (in some cases) after testing in 1 M KOH at 80 °C for 1000 h. PFTP-13 AEMs are stable in 1 M NaOH at 80 °C for 2000 h and maintain transparency and mechanical toughness after ex-situ durability testing (Fig. 6a and Supplementary Fig. 25). ~20% loss in ion conductivity was found in PFTP-13 after alkaline treatment in 5 M NaOH at 80 °C for 2000 h. Meanwhile, PFTP-13 AEMs were unstable under 10 M NaOH conditions, and significant degradation in chemical structure and mechanical loss were detected after 168 h (Supplementary Figs. 26–28).

As presented in Fig. 6b, in-situ durability testing of AEMFCs demonstrated that the present AEMFCs based on PFBP-14 ionomers and a PFTP-13 membrane can be operated stably under a 0.2 A cm$^{-2}$ current density at 70 °C in H$_2$–O$_2$ with a low A/C flow rate, and only 3.68% voltage decay (~130 μV h$^{-1}$) was observed after ~200 h. On the other hand, the commercial FAA membranes and ionomer-based AEMFCs showed significant voltage loss (over 40% voltage decay) within 48 h (Supplementary Fig. 29). Peng et al. reported 120 h in-situ durability at a 0.2 A cm$^{-2}$ current density (~10% voltage decay, ~800 μV h$^{-1}$) at 80 °C based on PFTP-0 AEMs[36]. Yan and coworkers[17] presented 300 h in-situ durability (~11.5% voltage decay at 250 h, ~300 μV h$^{-1}$) at a 0.5 A cm$^{-2}$ current density at 95 °C based on PAP-TP-$x$ AEMs. Very recently, a few state-of-the-art AEMFCs[3,46] reported ~1000 h in-situ durability (voltage decay 32–350 μV h$^{-1}$), and GT-$x$-based AEMFCs[2] could even obtain 2000 h in-situ durability (only 3.65% voltage decay, ~15 μV h$^{-1}$) under a 0.6 A cm$^{-2}$ current density at 75 °C.

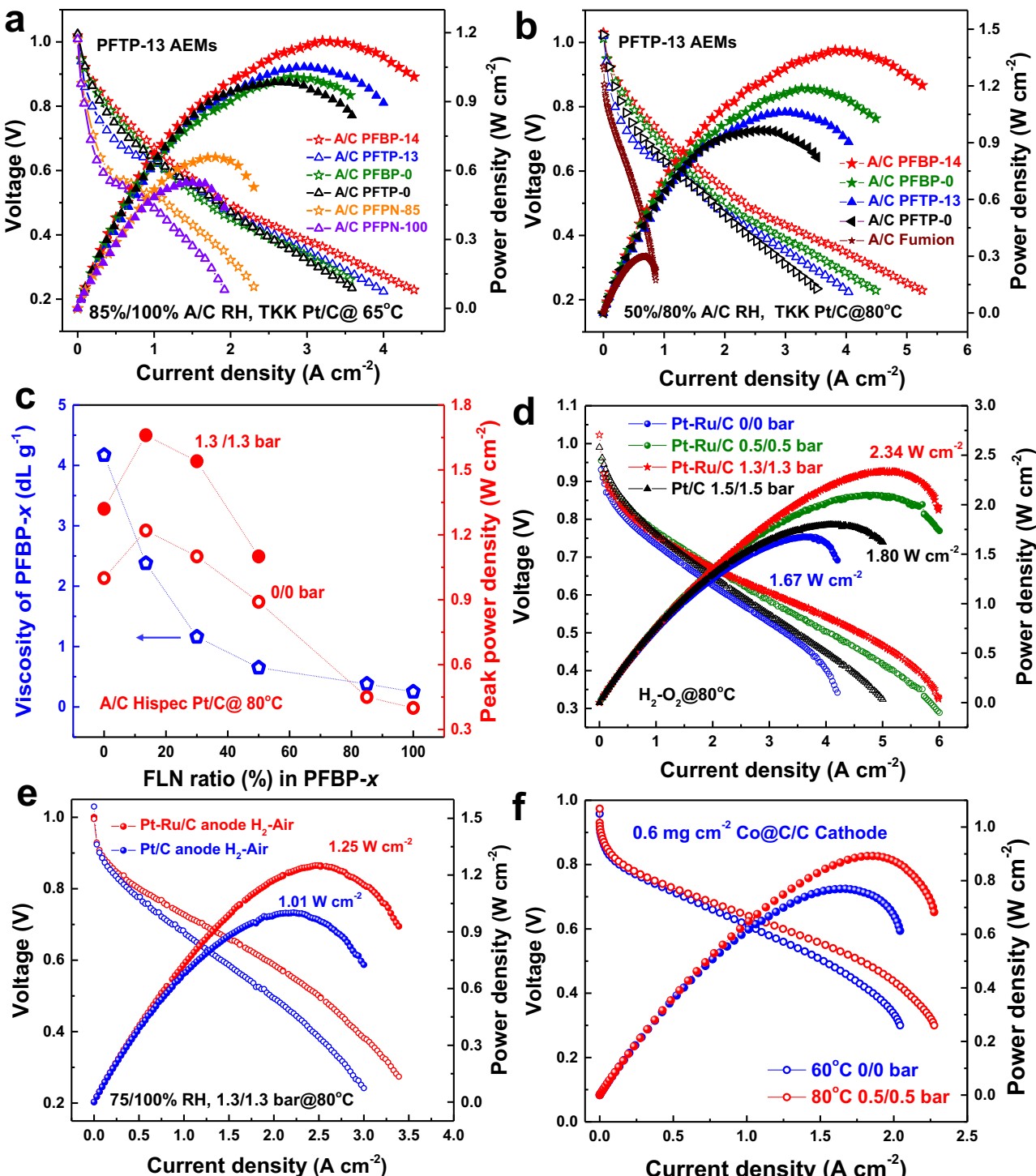

**Fig. 4 Fuel cell performance. a** The power density of AEMFCs with different AEIs and PFTP-13 AEMs (25 ± 3 μm) based on A/C TKK Pt/C catalysts (0.33 mg cm⁻²): 1000/1000 mL min⁻¹ H₂–O₂ flow rate, 65 °C, 85%/100% A/C RH, and 0.5/0.5 bar back pressure. **b** the power density of AEMFCs with different AEIs and PFTP-13 AEMs (25 ± 3 μm) based on A/C TKK Pt/C catalysts (0.33 mg cm⁻²): 80 °C, 50%/80% A/C RH, and 1.3/1.3 bar backpressure. **c** The relationship between the PPDs and the intrinsic viscosity of PFBP-x AEIs: PFTP-13 AEMs (25 ± 3 μm), H₂–O₂, A/C Hispec Pt/C catalysts (0.33 mg cm⁻²), 80 °C, the A/C flow rate of 1000/1000 mL min⁻¹, 75%/100% A/C RH. Hollow circle symbols are PPDs without back pressure, while filled circle symbols are PPDs with 1.3/1.3 bar back pressure. **d** The power density of AEMFCs based on Pt–Ru/C anode with backpressure: PFBP-14 AEIs and PFTP-13 AEMs (20 ± 3 μm), 80 °C, 75%/100% A/C RH, 1000/1000 mL min⁻¹ H₂–O₂ flow rate, different back pressures, Pt–Ru/C anode (0.42 mg cm⁻²), Hispec Pt/C cathode (0.33 mg cm⁻²). A/C Hispec Pt/C (0.33 mg cm⁻²) for comparison. **e** the power density of AEMFCs in H₂–air (CO₂ free) with different anode catalysts: PFBP-14 AEIs and PFTP-13 AEMs (20 ± 3 μm), 80 °C, 75%/100% A/C RH, 1000/2000 mL min⁻¹ flow rate, 1.3/1.3 A/C backpressure, 0.33 mg cm⁻² A/C catalyst loading. **f** PFAP-based AEMFCs with 0.6 mg cm⁻² loading of Co@C/C cathode and Pt–Ru/C anode with 1000/1000 mL min⁻¹ H₂–O₂ flow rate at 60 and 80 °C. PPDs reached 0.769 cm⁻² and 0.891 W cm⁻² at 60 °C without backpressure and at 80 °C with 0.5/0.5 bar backpressure, respectively.

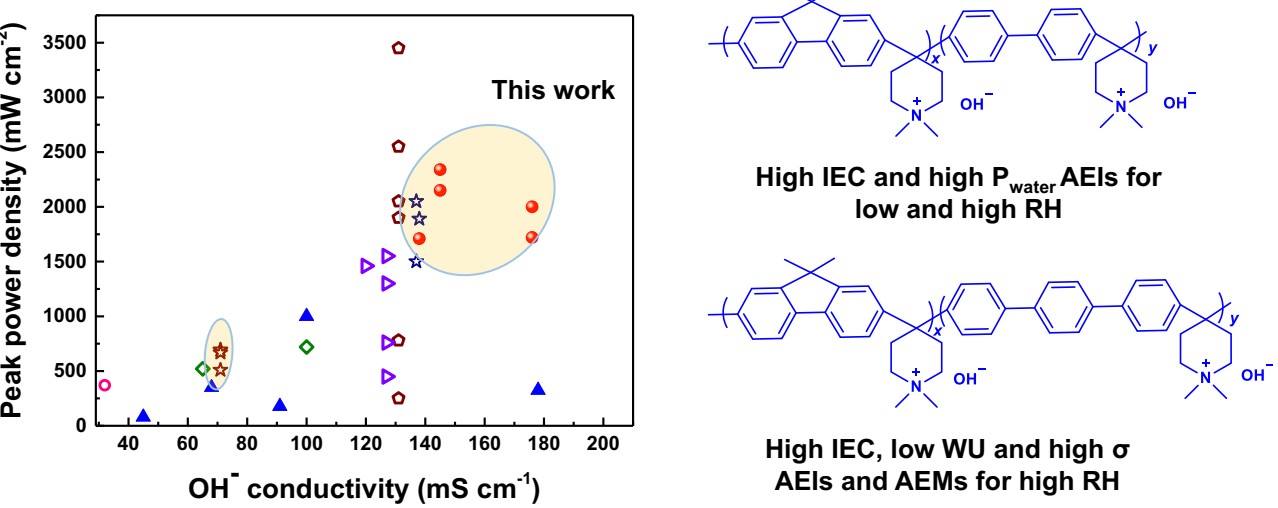

**Fig. 5 Comparison of representative AEIs in current research in regard to OH⁻ conductivity and PPDs at 80 °C.** PFAP (red cycle) and FAA Fumion ionomers (brown star) in this work. PSF/PPO ionomers (blue triangle symbols): BMTA-polysulfone[49], BTMA-PPO[11], DMP-PPO, ASU-PPO[26], and multication side chain PPO[50]. PBI ionomers (pink circle symbols): HMT-PBI[26]; BTMA-ETFE ionomers (brown pentagon symbols)[12,15,29-31,33]. BTMA-SEBS ionomers (green tetragon symbols)[51,52], PAP ionomers (blue star symbols): PFTP-0 and PFBP-0 ionomers[17,32,36], but the OH⁻ conductivity of PFBP-0 is currently missing. PF/PP ionomers (purple triangle symbols): side-chain polyfluorene and polyphenylene[34,37].

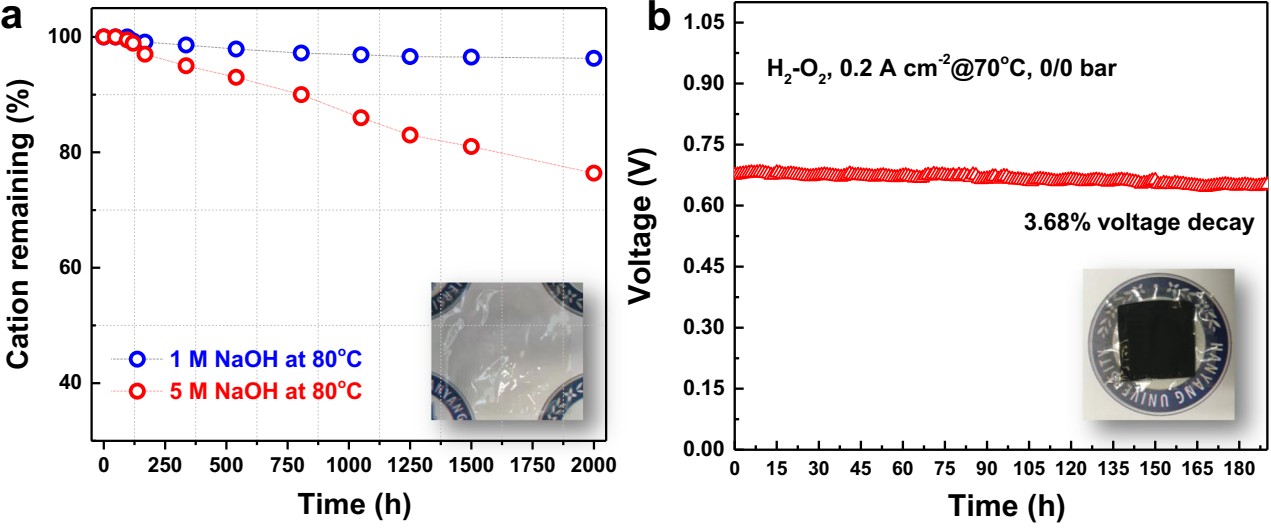

**Fig. 6 Ex-situ and in-situ durability. a** Remaining cations (%) of PFTP-13 AEMs at different concentrations of NaOH (1 and 5 M) at 80 °C for 2000 h detected by $^1$H NMR, along with the picture of membrane after alkaline treatment in 1 M NaOH for 2000 h. 3% and 22% loss in ion conducting groups (piperidinium) was detected by $^1$H NMR testing after immersion in those alkaline solutions. **b** In-situ durability of $H_2$–$O_2$ AEMFCs based on PFBP-14 AEIs and PFTP-13 AEMs testing at a 0.2 A cm$^{-2}$ current density at 70 °C with a 200/200 mL min$^{-1}$ A/C flow rate and 90/100% A/C RH, along with the picture of CCM after in-situ durability testing.

## Discussion

In summary, a series of high-performance PFAP-*x* AEIs and AEMs have been developed in this work. PFTP-13 AEMs simultaneously possessed >80 MPa TS and ~1500 MPa SM, excellent dimensional stability, and over 200 mS cm$^{-1}$ OH$^-$ conductivity at 98 °C. AEMFC performance demonstrated that PFBP-14 AEIs with suitable water vapor permeability exhibited superior PPDs at moderate or low RHs, which improved the mass transport efficiency and water diffusion in AEMFCs. The present AEMFCs reach PPDs of 2.34 and 1.25 W cm$^{-2}$ in $H_2$–$O_2$ and $H_2$–air, respectively. In-situ durability demonstrated that the present AEMFCs can be operated stably under a 0.2 A cm$^{-2}$ current density for ~200 h at 70 °C. All these results indicate that

these rigid PFAP AEMs and AEIs are promising candidates for AEMFCs.

## Methods

**Synthesis of poly(fluorene-co-aryl methylpiperidine) (PFAM-x).** A typical synthesis procedure of PF$_{0.13}$T$_{0.87}$M (or PFTM-13) is as follows: terphenyl (8.28 g, 36 mmol), 9,9′-dimethylfluorene (0.777 g, 4 mmol), and 1-methyl-4-piperidone (5.12 mL, 44 mmol) were added into a three-neck reactor, and then dichloromethane ($CH_2Cl_2$, 32 mL) was added to dissolve the monomers with mechanical stirring, followed by $N_2$ purge for 10 min. After the temperature of the solution was cooled to −3 °C by a chiller, trifluoroacetic acid (TFA, 4.8 mL) and trifluoromethanesulfonic acid (TFSA, 32 mL) were slowly added into the above solution. The color of the above solution immediately became dark red after adding TFSA. The reaction was kept at −3 °C by continuous mechanical stirring at ~10%

RH for 7–12 h depending on the viscosity of the solution. Subsequently, a very viscous polymer solution with a black-red color was carefully poured into a 1 M NaOH solution to produce white and stringy polymers. The polymer was smashed by a blender, and carefully washed several times with DI water until the pH was neutral. Finally, the polymer was dried in a vacuum oven at 80 °C to obtain a pale yellow PFTP-13. Yield: >95%. [1]H NMR (600 MHz, DMSO, δ): 7.21−7.70 ppm ($H_{e-h}$), 3.49 ppm ($H_a$), 3.15 ppm ($H_{a'}$), 2.83 ppm ($H_b$), 2.75 ppm ($H_c$), 2.31 ppm ($H_{b'}$), and 1.32 ppm ($H_d$) (Supplementary Fig. 1a).

A typical synthesis procedure of $PF_{0.14}B_{0.86}M$ (or PFBM-14) is as follows. Biphenyl (4.158 g, 27 mmol), 9,9′-dimethylfluorene (0.5828 g, 3 mmol), 1-methyl-4-piperidone (3.838 mL, 33 mmol), and $CH_2Cl_2$ (24 mL) were added into a three-neck reactor with mechanical stirring. After the temperature of solution was cooled to −3 °C, TFA (3.6 mL) and TFSA (24 mL) were slowly added into the above solution. The color of the above solution immediately became dark red after adding TFSA. The reaction was kept at −3 °C by continuous mechanical stirring at ~10% RH for 4–6 h depending on the viscosity of the solution. Subsequently, a viscous and dark red polymer solution was carefully precipitated in a 1 M NaOH solution. The polymer was carefully washed in DI water three times until the pH became neutral. Finally, the polymer was dried in a vacuum oven at 80 °C to obtain a pale yellow PFBM-14. Yield: >95%. [1]H NMR (600 MHz, DMSO, δ): 7.29 − 7.70 ppm ($H_{e-h}$), 3.50 ppm ($H_a$), 3.17 ppm ($H_{a'}$), 2.83 ppm ($H_b$), 2.75 ppm ($H_c$), 2.29 ppm ($H_{b'}$), and 1.32 ppm ($H_d$) (Supplementary Fig. 2).

**Synthesis of PFAP-x copolymers**. A typical synthesis procedure of $PF_{0.13}T_{0.87}P$ (or PFTP-13) is as follows: 4 g of $PF_{0.13}T_{0.87}M$ (or PFTM-13) was dissolved in 40 mL of DMSO at 80 °C with 1 mL of TFA as a cosolvent. After PFTM-13 was completely dissolved in DMSO, the polymer solution was cooled to room temperature. Then, 2.5 g of potassium carbonate and 2 mL of $CH_3I$ (3eq) were added to the above solution. The quaternization reaction was kept at room temperature for 24 h with a tinfoil covering to avoid light. After reaction, the polymer solution was precipitated in ethyl acetate to obtain a polymer, and then the polymer was filtered and washed twice with DI water to remove residual inorganic salt. Finally, the polymer was dried in a vacuum oven at 80 °C for 24 h, and 4.8 g of white PFTP-13 was obtained. Yield >90%. [1]H NMR (600 MHz, DMSO, δ): 7.5−7.85 ppm ($H_{e-h}$), 3.31 ppm ($H_a$), 3.12 ppm ($H_c$), 2.90 ppm ($H_b$), and 1.35 ppm ($H_d$) (Supplementary Fig. 1b). The synthesis procedure of PFBP-14 is similar to PFTP-13.

**Synthesis of poly(fluorene N-methylpiperidine-co-nonafluoride) copolymer (PFMN-x) and PFPN-x**. A typical synthesis procedure of PFMN-85 is as follows: 1.94 g of 9,9′-dimethylfluorene (10 mmol), 1.28 mL of 1-methyl-4-piperidone (0.98 mL, 8.5 mmol), 1,1,1,2,2,4,5,5,5-nonafluoro-4-(trifluoromethyl)pentan-3-one (0.296 mL,1.5 mmol), and 8.7 mL of $CH_2Cl_2$ were added into a three-necked reactor with mechanical stirring. TFA (1.2 mL) and TFSA (8 mL) were slowly added into the above solution at −3 °C. The color of the above solution immediately became dark red at the beginning, and the reaction was kept at −3 °C with continuous stirring for 6 h. Subsequently, the polymer solution was precipitated in a 1 M NaOH solution and was carefully washed several times in DI water until the pH was neutral. Finally, the polymer was dried in a vacuum oven at 80 °C to obtain a yellow PFMN-85. Yield: ~70%.

Then, 1 g of PFMN-85 was dissolved in 30 mL of DMSO to form a homogeneous solution. Next, 1 mL of $CH_3I$ (>3eq) was added to the above solution. The quaternization reaction was kept at room temperature for 24 h with a tinfoil covering to avoid light. After the reaction, the polymer solution was precipitated in ethyl acetate and was washed twice in DI water. Finally, the polymer was dried in a vacuum oven at 80 °C for 24 h. Yellow PFPN-85 powder was obtained. Yield~80%. [1]H NMR (600 MHz, DMSO, δ): 7.2−7.81 ppm ($H_{e-g}$), 3.31 ppm ($H_a$) 3.12 ppm ($H_c$), 2.85 ppm ($H_b$), and 1.35 ppm ($H_d$) (Supplementary Fig. 10).

**Membrane casting and ion exchange**. 1 g of PFTP-13 polymer was dissolved in 29 g DMSO to prepare a 3.33 wt% polymer solution. Subsequently, the polymer solution was collected into a syringe and filtered by a 0.45 μm filter, and then the solution was cast into a 14 × 21 cm glass plate. The polymer solution was dried in an oven at 90 °C for 24 h to slowly remove the solvents, and then a visible membrane was heated at 140 °C for another 12 h under vacuum to remove the solvents. Finally, the membrane was peeled off from the glass plate, resulting in a film with a thickness of 20 ± 5 μm.

PFTP-13 membranes were soaked in 1 M NaOH, 1 M NaCl, and 1 M NaCO₃, respectively, at 60 °C for 24 h for ion-exchange to OH⁻, Cl⁻, and $CO_3^{2-}$ forms, respectively. After ion exchange, the color of the PFTP-13 membrane became fainter in the Cl⁻ form and then colorless in OH⁻ and $CO_3^{2-}$ forms.

**[1]H nuclear magnetic resonance**. The chemical structures of AEPs were confirmed by [1]H NMR (VNMRS 600 MHz, Varian, CA, USA). d₆-DMSO was used as a solvent with a standard chemical shift of 2.50 ppm. Then, 10% TFA was added to [1]H NMR samples to eliminate the water peak effect (3.34 ppm).

**Solubility testing**. The solubilities of the polymers were measured in DMSO, DMF, NMP, and DMAc. Moreover, the solubility of the AEI solution (5% DMSO) was measured in isopropanol (IPA)/deionized (DI) water (10 to 1).

**IEC, WU, SR, and ion conductivity ($\sigma$)**. The IEC values of the AEIs were calculated by [1]H NMR through the relative integral area between the aromatic and methyl protons. The WU and SR of membranes were measured in OH⁻ and Cl⁻ forms. After ion exchange, a membrane in a specific form was washed with DI water several times, and then the hydrated membrane was wiped quickly using a filter paper to remove the surface water. The weight ($m_{wet}$) and unidirectional length ($L_{wet}$) of the wet membrane were recorded. Then, the membrane was dried in a vacuum oven to constant weight by covering it with a filter paper to avoid membrane shrinkage. Subsequently, the dry weight ($m_{dry}$) and the length ($L_{dry}$) of the membrane were recorded immediately. In-plane and through-plane SR were measured. WU and SR were calculated according to the following equations:

$$WU(\%) = \frac{m_{wet} - m_{dry}}{m_{dry}} \times 100\% \quad (1)$$

$$SR(\%) = \frac{L_{wet} - L_{dry}}{L_{dry}} \times 100\% \quad (2)$$

The ion conductivity of AEMs was measured using a four-probe method by an AC impedance analyzer (VSP and VMP3 Booster, Bio-Logic SAS, Grenoble, France) over the frequency range from 0.1 to 100 kHz. AEM samples in different forms were cut into 1 × 3 cm rectangular shapes (width = 1 cm), and then the membranes were fixed between two Pt wire electrodes in a fuel cell test station (CNL, Energy Co., Seoul, Korea). The distance ($L$) between the two electrolytes was 1 cm. The thickness ($d$) of the membrane sample was measured using a micrometer caliper. In-plane ion conductivity ($\sigma$) was measured at fully hydrated conditions (RH = 100%) at elevated temperatures, and the resistance ($R$) of the membrane was recorded. The ion conductivity was calculated from the following equation:

$$\sigma = \frac{d}{RLW} \quad (3)$$

Hydration number ($\lambda$), which represents the number of water molecular per OH⁻, was calculated using the following equation:

$$\lambda = \frac{WU \times 10}{IEC \times 18} \quad (4)$$

**Dynamic vapor sorption (DVS)**. The water sorption of AEM samples at different RHs was measured by a DVS (Surface Measurement Systems, UK) instrument at 25 °C. AEM samples were dried in a vacuum oven at 100 °C to remove residual water before testing. During testing, RH was automatically increased from 0% to 90% with six steps (0%, 18%, 36%, 54%, 72%, 90%), and then decreased from 90% to 0% step by step, and every RH stage was stable for 1 h to reach water equilibrium.

**DFT calculations**. Spin-polarized DFT calculations were performed for estimating the adsorption energy of 9,9-dimethylfluorene-dimethylpiperidinium (FL-DMP) and biphenyl-dimethylpiperidinium (BP-DMP) on the surfaces of Pt (111) and Pt–Ru (111) using the Vienna Ab initio Software Package (VASP) code, which is based on first principles. The Perdew–Burke–Ernzerhof (PBE) approach within the generalized gradient approximation (GGA) was adopted to examine the electronic exchange correlation function of the interacting electrons. Input parameters were set as follows: energy cutoff (400 eV), energy criteria ($10^{-5}$ eV), force criteria (0.05 eV/Å), smearing method (Methfessel-Paxton), and broadening width (0.2 eV). In this study, adsorption reactions were described as a large scaled unit cell ($a$: 22 Å, $b$: 22 Å, $c$: 30 Å) to avoid direct interactions between the periodic images. The calculated adsorption energy was determined from the equation below:

$$\triangle E_{ad} = E_{s+ad} - [E_s + E_{ad}] \quad (5)$$

Here, $\triangle E_{ad}$ denotes the adsorption energy of the molecule on the catalyst surface. $E_{s+ad}$ denotes the total energy of the adsorbed system. $E_s$ denotes the total energy of the catalyst, and $E_{ad}$ denotes the total energy of the adsorbates. Different adsorption directions of ionomers on the surface of catalysts were considered to compare the optimized adsorption energies of phenyl and ammonium groups.

**Torsional rotation calculation**. The structural models were constructed by Materials Studio 8.0 (Accelrys). Dihedral angles in the optimized geometry were determined by using the Dmol3 module (GGA/BLYP method and DND basis set). The relative energy variation of highlighted bonds with the torsional rotation angles was measured by the conformers package in Materials Studio 8.0. Energy minimizations of the structures of each conformer generated were done by using the Forcite Module (forcefield assigned COMPASS and Smart algorithm for iterations).

**Mechanical properties and thermal stability**. A universal testing machine (UTM, AGS-J 500N, Shimadzu, Japan) was used for measuring the mechanical properties of membrane samples. The TSs and elongations at break (EB) of PFBP-14, PFBP-0,

PFTP-13, PFTP-0, and commercial FAA-3-20 membranes in halogen form were measured in the dry state. All membrane samples were cut into a dumbbell shape ($2 \times 10$ mm), and the stretching rate was 1 mm/min. The thickness of the membranes was recorded using micrometer-scale calipers. In addition, the mechanical properties of PFTP-13 and PFTP-0 membranes in different forms and at different temperatures were also investigated. Moreover, the TS and EB of the PFTP-13 membrane after in-situ stability testing was investigated.

The thermal stabilities of AEPs were measured using a thermogravimetric analysis instrument (TGA, Q500, TA Instrument, USA) connected to a mass spectrometer (MS, ThermoStar™ GSD 301T, Pfeiffer Vacuum GmbH, Germany). The temperature was increased from 30 to 800 °C at a 10 °C/min ramping rate under a nitrogen atmosphere. Mass spectroscopy was used to detect the thermally decomposed species released from AEPs at related temperatures.

**Dynamic mechanical analysis (DMA)**. DMA (Q800, TA Instrument, DE, USA) was employed to measure the glass-transition ($T_g$) temperature, SM, and loss modulus (LM) of AEPs. Specifically, all membrane samples were cut into a $0.9 \times 2$ cm shape and were fixed with tension clamps in the DMA system. DMA testing was performed with a 1 Hz single-frequency strain mode, a preload force of 0.01 N and a force track of 125% under a nitrogen atmosphere. The target temperature was set to 450 °C along with a 10 °C/min ramping rate.

**Intrinsic viscosity**. The intrinsic viscosity ($[\eta]$) of AEPs was measured using a Ubbelohde viscometer in a DMSO solution at 25 °C. The viscometry system is composed of a Schott Viscosystem (AVS 370, Germany), Ubbelohde viscometer (SI Analytics, Type 530 13: Capillary No. Ic, $K = 0.03$) and piston buret (TITRONIC universal). The polymer solution was gradually diluted into five different concentrations, and the efflux time was automatically and repetitively recorded by the system five times. The reduced ($\eta_{red}$), inherent ($\eta_{inh}$), and intrinsic viscosities can be calculated using the following equations:

$$\eta_{red} = \left( \frac{t_1}{t_0} - 1 \right) / c \tag{6}$$

$$\eta_{inh} = \left( \ln \frac{t_1}{t_0} \right) / c \tag{7}$$

Here, $t_1$ is the efflux time of a polymer solution, $t_0$ is the efflux time for a DMSO solution, and $c$ is the concentration of the polymer solution. In a plot of $\eta$ versus $c$, the $y$-intercept was obtained by extrapolating the $\eta_{red}$ and $\eta_{inh}$ to $c = 0$. The intrinsic viscosity was obtained by calculating the average of the obtained $y$-intercept values.

**Differential scanning calorimetry**. DSC (Q20, TA Instrument, DE, USA) was employed to determine the number of free water molecules ($N_{free}$) and bound water molecules ($N_{bound}$) present on the AEMs in OH⁻ form. DSC analysis was performed with an aluminum pan under a 50 mL min⁻¹ nitrogen flow rate, and the heating temperature was gradually increased from −55 to 20 °C along with a 3 °C min⁻¹ ramping rate. $N_{free}$ and $N_{bound}$ were determined by the following equations:

$$\lambda = N_{free} + N_{bound} \tag{8}$$

$$N_{free} = \frac{H_f / H_{ice}}{(M_{wet} - M_{dry}) / M_{wet}} \times \lambda \tag{9}$$

Here, $H_f$ is the enthalpy obtained by the integration of the freezing peak calculated from the DSC program. $H_{ice}$ is the enthalpy of water fusion. $M_{wet}$ and $M_{dry}$ are wet and dry masses of a membrane sample, respectively.

$$H_{ice} = H^o_{ice} - \triangle C_p \triangle T_f \tag{10}$$

where $\triangle C_p$ is the difference between the specific heat capacity of liquid water and ice. $\triangle T_f$ is the freezing point depression.

**Gas permeability**. H₂, O₂, and water vapor permeabilities of PFTP-13, PFBP-14, PFTP-0, PFBP-0, commercial FAA-3-20 and Nafion 212 membranes were performed using a laboratory made gas permeability testing system (Supplementary Fig. 30) connected with a gas chromatograph (GC, 490 Micro GC, Agilent Technologies, USA) and two mass flow controllers (MFC, M3030V, Line Tech, Korea) operating at different RHs (from 0% to 90% RH) at 60 °C ($T$) under a 2.2 bar unilateral back pressure[47,48]. The gas permeability ($P$) can be calculated by the following equation:

$$P = \frac{V M_{gas} d}{P_{feed} R T A \rho} \frac{d_p}{d_t} \tag{11}$$

Here, $A$ (4.9 cm²) and $d$ (µm) represent the effective area and thickness of membrane samples, respectively. $P_{feed}$ and $M_{gas}$ (g mol⁻¹) are the pressures of each gas (760 mmHg) and molecular weight of the permeating gas, respectively. $V$ (cm³) is the volume of the measuring device at the bottom of the membrane samples. $\rho$ (g cm⁻³) and $R$ (L mmHg K⁻¹ mol⁻¹) are the densities of the permeating gas and the gas constant, respectively. $\frac{d_p}{d_t}$ is the slope that can be plotted from a change in

permeated gas pressure as a function of time. The unit of $P$ is Barrer where 1 Barrer = $10^{-10}$ cm³ (STP) cm cm⁻² s⁻¹ cm Hg⁻¹.

**Morphology analysis**. The surface and cross-section morphologies of AEMs and MEAs were observed using a scanning electron microscope (SEM, FE-SEM S-4800, Hitachi, Japan) at 15 kV. Membrane and MEA samples for observing the cross-section morphologies were fractured in liquid nitrogen, and all samples were coated with a thin platinum layer using an ion sputtering system (E-1045, Hitachi). Atomic force microscopy (AFM) was used to observe the surface microphase separation of AEMs on a MultiMode 8 AFM (Veeco) with a NanoScope V controller. AEMs were tested in OH⁻ form in the dry state.

**Fuel cell testing**. PFTP-13, PFTP-0, and commercial FAA-2-30 membranes were selected as AEMs. PFTP-13, PFBP-14, PFBP-0, PFTP-0, PFPN-100, PFPN-85, and FuMA-Tech Fumion ionomers were used as AEIs. Pt/C (Tanaka Kikdfinzoku Kogyo-TKK, 46.6 wt% Pt/C), Pt/C (Johnson Matthey HiSpec 4000, 40 wt% Pt/C), Pt–Ru/C (Johnson Matthey HiSpec 10,000, 40 wt% Pt and 20 wt% Ru) and home-made Co@C/C (40 wt% Co) were employed as catalysts. AEIs were dissolved in DMSO to prepare a 5% polymer solution, and then the polymer solution was filtered using a 0.45 µm PTFE filter. The catalyst slurry was prepared by adding the 5% AEIs/DMSO solution and catalysts into IPA/DI water (10 to 1), and then the slurry was sonicated for 1 h. Subsequently, the catalyst slurry was sprayed onto both sides of AEMs (I⁻ form) using an airbrush to produce 5 cm² CCM. 12.5 mg Hispec Pt/C, or 8.33 mg Pt–Ru/C, or 11.1 mg TKK Pt/C can produce the catalyst loading of $0.4 \pm 0.05$ mg cm⁻² in a dry CCM. The ratio of catalysts to ionomers is 3.33:1. CCMs were immersed in 1 M NaOH at room temperature overnight, and they were then washed with DI water three times. CCMs in the wet state were directly assembled with two gas diffusion layers (GDLs), fluorinated ethylene propylene (FEP) gaskets, and graphite bipolar plates with a 5 cm² flow field to obtain a complete AEMFC using a torque of 70 in-lb. The actual catalyst loading of MEAs was slightly adjusted by the area swelling of wet CCMs.

AEMFC performance was measured using a fuel cell test station (CNL, Seoul, Korea). Single cells were activated by scanning the current from 0 to 3 A cm⁻² with a 1000 mL min⁻¹ flow rate at 80 °C for two cycles without backpressure, and the polarization curve and power density curve were recorded after activation. Fuel cell performance was tested under different conditions, including temperature, RH, back pressure, flow rate, and feed gas.

**Ex-situ and in-situ durability**. The ex-situ durability of the PFTP-13 membrane was measured in 1, 5, and 10 M NaOH at 80 °C for 2000 h. The degradation ratio of PFTP-13 was calculated by the changes in chemical structure detected by ¹H NMR and Cl⁻ conductivity at room temperature. Changes in the mechanical properties and thermal stability were also investigated after alkaline treatment in 10 M NaOH at 80 °C for 168 h. Moreover, the in-situ durability was measured at a 0.2 A cm⁻² current density at 70 °C under H₂–O₂ conditions based on A/C PFBP-14 ionomers and a PFTP-13 membrane.

## Data availability

The data that support the plots within this paper and other findings of this study are available from the corresponding author upon reasonable request.

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

## Acknowledgements

This research was supported by the Technology Development Program to Solve Climate Change through the National Research Foundation of Korea (NRF) funded by the Ministry of Science and ICT (NRF-2018M1A2A2061979) and Material Component Technology Development (20010955) through the Korea Evaluation Institute of Industrial Technology (KEIT) funded by the Ministry of Trade, Industry and Energy, South Korea.

## Author contributions

Y.M.L. and N.J.C. conceived of the idea. N.J.C. wrote the draft. Y.M.L. guided the work and edited the manuscript. N.J.C. synthesized the polymers, fabricated the membranes, and measured ion conductivity and alkaline stability. N.J.C., H.H.W., and H.M.K. performed TGA and mechanical property measurements. H.H.W. and H.M.K. tested the intrinsic viscosity and performed DSC and DMA measurements. Y.-C.C. and E.S.S. performed the DFT calculations. H.H.W. performed SEM and TEM measurements. S.P.K. prepared MEA. N.J.C. and S.P.K. tested fuel cells and in-situ durability. C.H. and N.J.C. performed AFM measurements. W.H.L. and J.Y.B. tested water and gas permeabilities. Torsional rotation calculations were done by Y.Z. Co@C/C cathode-based AEMFCs were measured by S.J.Y. and J.-H.J.

## Competing interests

The authors declare no competing interests.
