## [Peer Review File · Nature Communications]

Reviewer #1 (Remarks to the Author):

In this work, the authors synthesis new polymers for HEMFCs. They report very high performance and the membranes seem to have good chemical stability. I have several comments that are suggested to be addressed:

1. The authors start the paper with references 1-3 by pointing out that HEMFCs have made significant advances in power density, but they do not reference any relevant work there where in the past 3 years the peak power has gone up significantly. In particular, they need to pay attention and reference the work by Y. Yan, L. Zhuang, W. Mustain, J. Varcoe and P. Kohl here at the very least. Some of these come later in the opening paragraph, but not in the proper context in my view.

2. They also need to look at recent work by D. Dekel on HEMFC stability and the new work by P. Kohl & W. Mustain in this journal, Nature Communications, showing 2000 hour very stable operation and high performance ($> 3\text{W}/\text{cm}^2$) with all poly-norbornene components.

3. The authors talk about water uptake and water transport as if they are interchangeable, but that does not appear to be the case. Uptake kinetics are not exactly representative of diffusivity, which will dominate once the film is fully hydrated when liquid water is produced in the cell. Such discussion might be helpful with the ionomers, but this reviewer is not sure that the team here has actually shown faster transport, and not under relevant conditions – particularly the permeability of liquid water under very close to fully hydrated conditions.

4. The abbreviations that are used in the main text, but defined in the Methods section, should be defined in the main text and then used as abbreviations in the Methods section.

5. The authors attribute the higher performance with back-pressurization to higher water content/transport. However, it could also be better access to porosity or higher reactant gas pressure. It is recommended that the authors do some experiments with similar gases partial pressures (both H_2 and O_2) and then compare.

6. On page 14, the authors compare their results to truly uninteresting cells. They say, “The PPD of the commercial Fumion ionomer and FAA-3-20 AEMs are much lower than the present PFAP AEMFC”. But these are not the state of art. These need to be compare to the work by the authors listed in the first point above.

7. Also, with regards to comparison, the highest work in the literature by P. Kohl et al is collected at 1atm total pressure, with no backpressure. That compares to the black curve in Figure 4d, which is around $\frac{1}{2}$ his value. In general, this paper lacks acknowledgement of where their work is relative to state of art – though it is quite good.

8. It is good that the authors show minimal polymer degradation during the lifetime testing, but the performance degradation suggests that their operation conditions are not good for the cell. Why is this? Is the membrane here not as water permeable as they suggest, even at such low current density? The authors should also explicitly give the RH's in Figure 5c.

9. The claim in the paper that the specific power is the highest value in the literature is completely not true. This reviewer knows of at least 1 work by W. Mustain group with Pt-free cathodes also that

show higher specific power than reported here.

Reviewer #2 (Remarks to the Author):

This manuscript reports poly(fluorenyl aryl piperidinium)s for anion exchange membrane fuel cells (AEMFCs). AEMFCs is an alternative energy conversion device that has been investigated over the past decade. Recent reports showed significant performance improvements of AEMFCs but the durability of AEMFCs remains the primary challenge for a commercially viable system. This manuscript discusses mostly the performance of AEMFCs using poly(fluorenyl aryl piperidinium)s. Poly(piperidinium)s have been made from Professor Jannasch's research group at Lund University, then Professor Yan's group at University Delaware showed high performance of AEMFCs employing a poly(piperidinium). The authors have modified the structure of the poly(piperidinium) by adding fluorene moiety to improve fuel cell performance. Fluorene is well-known fragment that showed high performance due to the low phenyl adsorption characteristics (see Maurya et al. Energy & Environ. Sci. Ref. 34). The authors combined the poly(piperidinium) with fluorene moiety which I think a clever idea, although the novelty of the manuscript is questionable.

The strength of this manuscript is extensive experimental works and PI has the right direction to carry out to develop polymer electrolytes. The fuel cell performance is impressive, I believe one of the highest in the polyaromatics-based AEMFCs. I am not sure it is the best one as the research focuses on the research by other research groups were different.

There are several weaknesses to this manuscript.

First, the durability of the AEMFCs is not high. The authors stated that this is due to water management but this does not give much insight to the readers. It is well known that AEMFC has a water management issue but not many people discuss what is the origin of water management.

Second, the authors discuss too many characteristics of AEMs and ionomers including conductivity, gas permeability, adsorption, molecular weight, water management. Thus, at the end of reading the manuscript, it is not clear which one is the major factor that brought high performance. If the authors believe that little bit of everything (I personally do not believe that), then the authors should give some idea how much of each contribution. For example, if hydroxide conductivity by fluorene moiety is the major factor, then the authors need to measure the cell resistance of several AEMs at the same thickness. If gas permeability is the major contribution, then authors should demonstrate the OCV of different cells to prove the impact.

Third, the authors indicated that PFPN-x polymers did not perform well in spite of low phenyl adsorption. The authors speculated that low performance is due to the insufficient molecular weight. I do not know this is true or not because the authors did not prove much of property data of PFPN-x (conductivity, mechanical property, water uptake). Moreover, the authors did not provide any molecular weight effect on performance. I do not believe, PFPN-x polymers are ready to publish in this manuscript.

Fourth, Figure 4 shows all ionomer effect since the same membrane PFTP-13 AEMs are used. The conclusion is reasonable that the cell using PFBP-14 AEMs performed well. However, if I read the main characterization part, PFTP-13 has more desirable properties. The liquid water uptake of PFBP-14 is

very high which may result in electrode flooding. Water permeability of PFBP-14 is high which is good for electrode but high water permeability is also required for AEM. The high water permeability PFBP-14 is probably due to high liquid water uptake (then again it is not a good property due to potential flooding issue). H₂ permeability of PFBE-14 is much lower than PFTB-13, which is a significant disadvantage as an ionomer. Hydroxide conductivity of PFBP-14 is also lower than PFTP-13. I agree that high mechanical properties are required more with AEM but PFTP-13 has better mechanical properties than PFBP-14. So it seemed to yield that most properties of PFBP-14 are not desirable compared to the PFTP-13 AEM, then suddenly the best performance was achieved with PFBP-14 in Figure 14. I do not find any electrochemical half-cell data of PFBP-14 so it is puzzling why PFBP-14 performs the best.

This manuscript can be published in Nature Communications without an in-depth study on the durability portion (the first item). However, the authors may need to address the other weakness (No.2 – 4) before submitting the revised manuscript.

RESPONSE TO REVIEWERS

Manuscript Number: NCOMMS-20-39303-T

Title: Durable poly(fluorenyl aryl piperidinium) membranes and ionomers for anion exchange membrane fuel cells

Authors: Nanjun Chen†, Ho Hyun Wang†, Sun Pyo Kim, Hae Min Kim, Won Hee Lee, Chuan Hu, Joon Yong Bae, Eun Seob Sim, Yong-Chae Chung, Jue-Hyuk Jang, Sung Jong Yoo, Yongbing Zhuang, Young Moo Lee*

Authors appreciate Editor and Reviewers for their time and efforts to provide critical comments to improve the present manuscript. We tried our best to answer all the questions from Reviewers as can be found in the following responses written and marked in blue. All the changes are made in the revised version marked in yellow. Marked in red below is the newly added sentences in the revised version.

Reviewer #1 (Remarks to the Author):

In this work, the authors synthesis new polymers for HEMFCs. They report very high performance and the membranes seem to have good chemical stability. I have several comments that are suggested to be addressed:

1. The authors start the paper with references 1-3 by pointing out that HEMFCs have made significant advances in power density, but they do not reference any relevant work there where in the past 3 years the peak power has gone up significantly. In particular, they need to pay attention and reference the work by Y. Yan, L. Zhuang, W. Mustain, J. Varcoe and P. Kohl here at the very least. Some of these come later in the opening paragraph, but not in the proper context in my view.

Response: Thank you for your suggestion. Researchers you listed are important people in AEMFCs field, and we paid a lot of attention to them and their publications. We cited the latest important papers published by Mustain *et al.* and Kohl *et al.* (*Adv. Energy Mater.* 2020, 10, 2001986. in **ref 2**) and *Nat. Commun.* 2020, 11, 3561 was cited in **ref 3** reporting excellent AEMFC durability. Also, we deleted some improper references (original ref 2 and ref 3) and modified the position of some references according to reviewer's suggestion.

2. They also need to look at recent work by D. Dekel on HEMFC stability and the new work by P. Kohl & W. Mustain in this journal, Nature Communications, showing 2000 hour very stable operation and high performance ($> 3\text{W}/\text{cm}^2$) with all poly-norbornene components.

Response: These are excellent papers related with *in-situ* durability of AEMFCs. We cited Dekel. *et al.* (*Adv. Funct. Mater.* **30**, 2002087 (2020) as ref 4, *Nat. Commun.* 2020, 11, 3561 as ref 3, and *Adv. Energy Mater.* 2020, 10, 2001986 as ref 2, which are highlighted in yellow in the revised manuscript.

3. The authors talk about water uptake and water transport as if they are interchangeable, but that

does not appear to be the case. Uptake kinetics are not exactly representative of diffusivity, which will dominate once the film is fully hydrated when liquid water is produced in the cell. Such discussion might be helpful with the ionomers, but this reviewer is not sure that the team here has actually shown faster transport, and not under relevant conditions—particularly the permeability of liquid water under very close to fully hydrated conditions.

Response: Thank you for your constructive suggestion. We agree with reviewer’s opinion that water uptake and water transport (represented as diffusivity or permeability) of AEIs or AEMs are not interchangeable. *High water uptake represents the water sorption capacity of AEIs or AEMs when liquid water is forming in AEMFCs, but does not represent their water transport.* Note that most of the polymer membranes follow the solution-diffusion mechanism, i.e., $P=D.S$ where P is the permeability, D=diffusivity and S=solubility coefficient. Water permeability measurement in this work was not conducted in fully hydrated conditions but we tried our best to measure it close to fully hydrated conditions. Solubility term is related with water sorption in equilibrium.

In addition to the reported water permeability in Figure 2d, we calculated the water diffusivity of these AEMs at different RH conditions based on DVS data, as shown in Table R1 (presented in Supplementary Table 4). PFBP-13 and PBP AEIs with high WU and IEC show the higher water diffusion coefficients than those of PTP and PFTP-13. Moreover, FLN-based AEIs—PFBP-14 or PFTP-13—display higher water diffusivities than those of PBP or PTP, respectively. We can see the similar trends between water diffusion coefficients and water permeability among the samples at different RH conditions.

In fully hydrated or liquid water conditions, the water diffusivity of AEMs could be much higher than at low RHs according to Dekel *et al.*’s report (*Journal of Membrane Science*, 2020, 608, 118206). Having said that, the water diffusion behavior of our AEIs is very similar to the reported Nafion series (*Journal of Power Sources*, 2006, 160, 426-430) or BPSH series (*Nature*, 2016, 532, 480-483). Basically, the water diffusivity increases fast at low RH, and then slows down at high RH due to the hysteresis phenomenon. This is a quite natural behavior of AEMs or PEMs at different RHs.

Related discussion is added in page 8 in manuscript, as follows:

“Notably, WU represents the water sorption capacity of AEMs or AEMs when liquid water is forming in AEMFCs, while it cannot represent the water transport behaviour. Therefore, water diffusivity of AEIs at different RHs was automatically estimated by DVS, as shown in Supplementary Table 4. PFBP-14 and PBP AEMs with higher IEC values exhibited higher water diffusivities than those of PFTP-13 and PTP AEMs. PFBP-14 and PFTP-13 AEMs also display higher water diffusivities than those of PBP and PTP AEMs, respectively, due to the presence of FLN blocks.”

Table R1. Water diffusivity of PBP, PFBP-14, PTP, and PFTP-13 AEMs in OH⁻ form at different RHs measured at 25 °C.

RH	Water diffusivity (10 ⁻⁷ cm s ⁻¹)			
	PBP	PFBP	PTP	PFTP-13

18 % RH	1.33	1.45	0.43	0.51
36% RH	1.43	2.08	0.67	1.13
54% RH	3.45	3.79	0.95	1.94
72% RH	2.32	-	1.36	2.53
90% RH	2.15	2.73	1.24	1.87

For reviewer’s query on our water permeability measurement in fully hydrated condition, actually, we focused on the water vapor permeability of AEIs at different RHs (0%, ~20%, ~40%, ~60%, ~80%, to ~90%). It is difficult to reach 100% RH in our permeability testing set-up due to the humidity control system. As depicted in the Supplementary Scheme 2 in Supporting Information, the humidity is controlled by mixing dry and wet gas at the feed side. Even if only wet gas was injected at the feed side of the membrane, the maximum RH was around 85% to ~90%. Therefore, we could not measure the water vapor permeability of AEIs in 100% RH, while the water permeability of different AEIs or AEMs should exhibit increasing water permeability at 100% RH compared to low RHs, and show similar tendency as well.

The gas permeability measurement system in this study is a lab-made apparatus originated from the testing system of gas separation membranes, covering the gas permeability of dry gases such as H₂, O₂, CH₄, N₂, etc. For this research we adjusted the system to measure the water permeability. The measurement conditions of the new system are close to actual AEMFCs, such as the system control of back pressure, temperature, and flow rate, etc. For water vapor permeability testing, only few groups can accurately measure it at different RHs. We believe this measurement is important to demonstrate water transport in AEMFC system with different RHs, and this new insight and approach can give clear-eyed guidelines in AEMFC areas in the future.

4. The abbreviations that are used in the main text, but defined in the Methods section, should be defined in the main text and then used as abbreviations in the Methods section.

Response: Thank you for your suggestion. We modified the abbreviations according to reviewer comments, which were highlighted in the revised manuscript.

5. The authors attribute the higher performance with back-pressurization to higher water content/transport. However, it could also be better access to porosity or higher reactant gas pressure. It is recommended that the authors do some experiments with similar gases partial pressures (both H₂ and O₂) and then compare.

Response: Thank you for your suggestion. We agree with the reviewer’s opinion that AEMFCs exhibiting high performance with back pressure can be attributed to high reactant gas pressure and improving water transport as well. We tested the AEMFC performance based on A/C Hispec Pt/C under different back pressures according to reviewer’s suggestion. Figure R1 below clearly shows the effect of back pressure on the power density of AEMFCs. A small amount of initial back pressure (0.5/0.5 bar) displays significant effect on the power density of AEMFCs due to high reactant gas pressure that increases HOR and ORR reactions. Besides, high back pressure also increases the water permeability of AEIs or AEMs (so-called water transport), so to optimize the water management and decrease the mass transport resistance.

This data is presented in supporting information as Figure S23b. The related discussion is provided in page 25 in Supporting Information as follows:

“On the other hand, the effect of back pressure on AEMFCs was investigated based on the PFTP-13 membrane and PFBP-14 A/C AEIs. Supplementary Figure 23b implies that a small back pressure shows a significant effect on the power density of AEMFCs. Based on Hispec Pt/C, the PPDs of AEMFCs increase from 1.08 W cm^{-2} to 1.80 W cm^{-2} at 80°C after gradually increasing A/C back pressure from 0 bar to 1.5 bar. Specifically, high back pressure increases the reactant gas pressure which results in higher electrode reaction rates. Moreover, high back pressure also contributes to improving the water permeability of AEMs or AEIs, which decreases the mass transport resistance.”

Figure R1. AEMFC performance based on the PFTP-13 membrane and A/C PFBP-14 ionomers with A/C Hispec Pt/C at 80°C in $\text{H}_2\text{-O}_2$ with $1000/1000 \text{ mL min}^{-1}$ A/C flow rate with different back pressures.

6. On page 14, the authors compare their results to truly uninteresting cells. They say, “The PPD of the commercial Fumion ionomer and FAA-3-20 AEMs are much lower than the present PFAP AEMFC”. But these are not the state of art. These need to be compare to the work by the authors listed in the first point above.

Response: Thank you for your comments. Many researchers are comparing their data with the commercially available AEMs and AEIs so that we have also compared our data with the commercially available FAA membranes and ionomers.

We carefully compared our data with GT-x series and BTMA-HDPE AEMs reported in refs. 2-4 according to reviewer’s suggestion. The related description is shown in page 15 in the revised manuscript.

“Currently, the state-of-the-art PTFE-reinforced PNB (GT-x series: 3.2 W cm⁻² in H₂-O₂ and 1.75 W cm⁻² in H₂-air without back pressure²; 3.5 W cm⁻² in H₂-O₂ and 1.25 W cm⁻² with 0.5/1.0 bar back pressure³¹) and BTMA-HDPE (2.35~2.5 W cm⁻² in H₂-O₂ and 1.06 W cm⁻² in H₂-air without back pressure)^{3,29} AEMs lead the current AEMFCs, while our present AEMFCs display comparable power density (2.34 W cm⁻² in H₂-O₂ and 1.25 W cm⁻² in H₂-air with 1.3/1.3 bar back pressure). On the other hand, compared to the state-of-the-art polyaromatics-based AEMFCs (<2.08 W cm⁻² in H₂-O₂ with 2.0/2.0 bar back pressure),^{17,32,34} our present AEMFCs showed outstanding PPDs.”

7. Also, with regards to comparison, the highest work in the literature by P. Kohl et al is collected at 1atm total pressure, with no backpressure. That compares to the black curve in Figure 4d, which is around ½ his value. In general, this paper lacks acknowledgement of where their work is relative to state of art – though it is quite good.

Response: Thank you for your suggestion. Kohl *et al's* AEMs or AEIs (GT-x series) are excellent materials in current AEMFCs, and the GT-x AEMs leading the power density and durability of AEMFCs so far. We appreciated for thir contribution in AEMFCs. Therefore, we added some related descriptions of these excellent AEMs in page 15, as follows:

“Currently, the state-of-the-art PTFE-reinforced PNB (GT-x series: 3.2 W cm⁻² in H₂-O₂ and 1.75 W cm⁻² in H₂-air without back pressure²; 3.5 W cm⁻² in H₂-O₂ and 1.25 W cm⁻² with 0.5/1.0 bar back pressure³¹) and BTMA-HDPE (2.35~2.5 W cm⁻² in H₂-O₂ and 1.06 W cm⁻² in H₂-air without back pressure)^{3,29} AEMs lead the current AEMFCs, while our present AEMFCs display comparable power density (2.34 W cm⁻² in H₂-O₂ and 1.25 W cm⁻² in H₂-air with 1.3/1.3 bar backpressure).”

Moreover, without back pressure, we can reach the PPDs to ~1.6 to ~1.8 W cm⁻² based on Pt-Ru/C, so we polish the data in Figure 4d in the revised manuscript.

8. It is good that the authors show minimal polymer degradation during the lifetime testing, but the performance degradation suggests that their operation conditions are not good for the cell. Why is this? Is the membrane here not as water permeable as they suggest, even at such low current density? The authors should also explicitly give the RH's in Figure 5c.

Response: Thank you for reviewer's suggestion. We specified the RH condition in Figure 5c, we used 90/100% A/C RH for our AEMFCs.

The limited durability of our AEMFCs is truly assigned to our improper or un-optimized operation conditions. Our target is to operate the AEMFCs with realistic conditions, such as employing low H₂ flow rate and low RH. However, we found the graphite bipolar plate and carbon papers were easy to be corroded during the durability testing, especially in the anode side, while AEMs were no problem after testing.

Normally, our AEMFCs can be operated under low flow rate for 100 h to 200 h at 0.4 A cm⁻² without MEA refreshment.

Having said that, our PFTP AEMs without any reinforcement or support showed the similar cell durability in PAP family without MEA refreshment, such as L. Zhuang (120 h, *Journal of Power Sources* 390 (2018) 165–167), Y. S. Kim (<100 h, *J. Mater. Chem. A*, 2020, 8, 14135, although they can finally reach 1000 h after MEA refreshment) and a little bit lower durability than Y. S. Yan’ AEMs (300 h, *Nature Energy*, 2019, 4(5), 392-398.)

Encouragingly, we have very impressive progress in recent durability test, and we could reach much high durability after addressing operating condition issues, while these data are not ready for publishing so far, which will be shown in our future publication.

9. The claim in the paper that the specific power is the highest value in the literature is completely not true. This reviewer knows of at least 1 work by W. Mustain group with Pt-free cathodes also that show higher specific power than reported here.

Response: Thank you for reviewer’s comment. Actually, we discussed the specific power of PGM-based AEMFCs (Pt/C or Pt-Ru/C). We found two papers related with PGM-free cathode and specific power published by Mustain *et al.* (*Angew. Chem.* 2019, 131, 1058-1063, *Journal of The Electrochemical Society*, 165 (9) F710-F717 (2018)) for comparison. We agree that AEMFCs with PGM-free cathode showed higher specific power due to the absence of Pt group. Related discussion is revised as follows:

“the SP in this work is outstanding (5.57 to 6.52 W mg⁻¹) in platinum group metal (PGM)-based AEMFCs, and the SP of BTMA-HDPE and GT series-based AEMFCs was between 4 to 5 W mg⁻¹. Compared to AEMFCs with PGM-free cathode^{44,45} (6 W mg⁻¹ to 8 W mg⁻¹), our present AEMFCs still showed comparable SP.”

Reviewer #2 (Remarks to the Author):

This manuscript reports poly(fluorenyl aryl piperidinium)s for anion exchange membrane fuel cells (AEMFCs). AEMFCs is an alternative energy conversion device that has been investigated over the past decade. Recent reports showed significant performance improvements of AEMFCs but the durability of AEMFCs remains the primary challenge for a commercially viable system. This manuscript discusses mostly the performance of AEMFCs using poly(fluorenyl aryl piperidinium)s. Poly(piperidinium)s have been made from Professor Jannasch’s research group at Lund University, then Professor Yan’s group at University Delaware showed high performance of AEMFCs employing a poly(piperidinium). The authors have modified the structure of the poly(piperidinium) by adding fluorene moiety to improve fuel cell performance. Fluorene is well-known fragment that showed high performance due to the low phenyl adsorption characteristics (see Maurya *et al.* *Energy & Environ. Sci.* Ref. 34). The authors combined the poly(piperidinium) with fluorene moiety which I think a clever idea, although the novelty of the manuscript is questionable.

The strength of this manuscript is extensive experimental works and PI has the right direction to carry out to develop polymer electrolytes. The fuel cell performance is impressive, I believe one of the highest in the polyaromatics-based AEMFCs. I am not sure it is the best one as the research focuses on the research by other research groups were different.

Response: Thank you for your positive comments in our manuscript.

There are several weaknesses to this manuscript.

First, the durability of the AEMFCs is not high. The authors stated that this is due to water management but this does not give much insight to the readers. It is well known that AEMFC has a water management issue but not many people discuss what is the origin of water management.

Response: Thank you for your comments. Actually, the reason why our cells cannot reach durability over 500 h or 1000 h is due to the improper and un-optimized operating conditions, such as carbon corrosion, cell refreshment, and improper GDL and MEA design. We could reach much higher cell durability after addressing these issues, while it is not ready to publish right now. Anyway, the limited durability of the present AEMFCs is not due to these PFAP AEMs and AEIs.

As reviewer pointed out, although water management of AEMFCs is well-known right now, while only a few researchers discuss the reasons. Since our target is to operate AEMFCs with realistic and economic conditions for future applications, therefore, we used low hydrogen flow rate to test the durability, which might be flooding after long-term testing. We thought the improper water management causes the electrochemical oxidation of carbon since we found that our graphitic bipolar plate and GDL were corrosion after durability testing. Moreover, the thickness, area, and imposed pressure of GDL, gasket, and AEMs are not well matched in our present MEA.

We changed the related description in page 19 and page 20 in the revised manuscript, intending to provide much insight for readers, as follows, and we cited a related reference here (*Leonard et al. J. Mater. Chem. A*, 2020, 8, 14135).

“Meanwhile, the voltage loss of the present cells is similar to Leonard et al.’s⁵⁰ cells within 100 h, although they can finally reach 1000 h *in-situ* durability after refreshing their MEA. The durability of the present cells is limited by improper and un-optimized operating conditions, such as improper water management and the electrochemical oxidation of carbon, since the corrosion of graphite bipolar plate and carbon papers was observed in our cells after durability testing. The condition optimization is our ongoing work, such as new MEA design and MEA refreshment.”

Second, the authors discuss too many characteristics of AEMs and ionomers including conductivity, gas permeability, adsorption, molecular weight, water management. Thus, at the end of reading the manuscript, it is not clear which one is the major factor that brought high performance. If the authors believe that little bit of everything (I personally do not believe that), then the authors should give some idea how much of each contribution. For example, if hydroxide conductivity by fluorene moiety is the major factor, then the authors need to measure the cell resistance of several AEMs at the same thickness. If gas permeability is the major contribution, then authors should demonstrate the OCV of different cells to prove the impact.

Response: Thank you for reviewer’s constructive comments, and this comment is related with Comment 4 below. We provide a detailed explanation how to obtain high AEMFC performance in the Comment 4.

In our opinion, the following three aspects are the key points to reach super high performance for current AEMFCs.

(1) Finding an excellent AEM with high mechanical property, ion conductivity, and reasonable WU are certainly the most important factors. (2) Finding an efficient ionomer with high ion conductivity, water permeability, reasonable WU, and low ionomer adsorption are another key point. Of course, the reasonable dimensional stability of ionomer is required. (3) Since the ORR and HOR reactions will be much faster when AEMFCs are operating **in high current density**, along with faster water consumption in the cathode. Therefore, the cathode dry-out and the anode flooding will dramatically limit the PPDs of AEMFCs particularly in high current density region. Therefore, based on our AEMs and AEIs, the key point to reach super high PPDs is to combine the high WU and water permeable AEIs with suitable anode and cathode RHs. We used 75/100% A/C RH for PFBP AEIs, which can well tackle the anode flooding issue, maintain the water content in the cathode, and avoid the AEI degradation at high current density. The related description is shown in page 14 in the revised manuscript, as follows:

“However, at high RH and moderate current density ($\sim 3.5 \text{ A cm}^{-2}$) (Fig. 4a) or at low RHs (50%/80%) and high current density ($> 5 \text{ A cm}^{-2}$) (Fig. 4b), PFBP-14 (1.42 W cm^{-2}) and PBP (1.19 W cm^{-2}) with high WU and P_{water} exhibited much higher PPDs than those of AEIs with low WU and P_{water} (PFTP-13: 1.06 W cm^{-2} , PTP: 0.98 W cm^{-2}). Note that the PPDs of PFTP-13 and PTP AEIs with low P_{water} cannot be further improved at high temperature with low RH due to severe electrode dry-out.”

“Obviously, at high current density, the electrode reactions were dramatically increased along with rapid water consumption in the cathode, resulting in the cathode dry-out (anode flooding as well). The high WU and P_{water} of AEIs contribute to maintaining the water content in the cathode and improving the water back diffusion at high current density or low RH conditions. However, employing high WU AEIs may cause the anode flooding in high current density. Therefore, after repetitive verification, we demonstrated that 75%/100% A/C RH for high WU and P_{water} PFBP-14 ionomer can well tackle the anode flooding issue, maintain the water content in the cathode, and avoid the AEI degradation at high current density. This is a key strategy to obtain super high PPDs in the present AEMFCs. The detailed optimization will be presented in our future work.”

Based on this strategy and the PFBP-14 ionomer, our PPD can reach over 2.6 W cm^{-2} so far after adjusting the water content of AEMs and further tackling the flooding in the anode.

We are sorry that we cannot provide the detailed contribution of each part in electrochemical performance because it is too complex to simply calculate each contribution. Of course, the high ion conductivity of AEMs is very important, contributing to the low internal resistance, as shown in Figure R2 (this data is presented in Supplementary Figure 18b), while we only used one type of AEMs (PFTP-13) here. As for the ion conductivity of AEIs, compared to AEMs ($> 5 \text{ cm}^2$, $\sim 25 \text{ }\mu\text{m}$, $50\sim 100 \text{ mg}$), we think AEIs possessing sufficient ion conductivity is good enough since only a small amount of ionomers ($\sim 3.75 \text{ mg}/3.75 \text{ mg A/C}$ (a total of 7.5 mg)) were added into catalyst layer. Therefore, the high-water permeable ionomers with matching RH is much more important factor. On the other hand, the low ionomer adsorption is truly important since only PFBP ionomer can reach super high PPDs with Pt-Ru/C, while PBP and PTP ionomers never go that far in our case.

As for fuel gas permeability of ionomers, H_2 and O_2 permeabilities of PFBP ionomers and PFTP membrane are not very high as permeabilities are reported to be less than 10 Barrer, while those of Nafion 112 and 212 or even PAP-TP-85 membranes are between 30-90 Barrer at different RHs. It is a general idea that AEIs should have high fuel gas permeability but actually PFBP ionomers with low H_2 and O_2 permeability display outstanding fuel cell performance with normal OCV (~ 1

V after activation). We checked different types of AEMFCs, and we cannot find too many information from OCV based on Pt/C, as shown in Table R2. Generally, the OCV of our AEMFCs is higher than 1.0 V after activation or cell testing. Based on Pt-Ru/C, the initial OCV is usually around ~0.9 V, and over 0.95 V after cell testing.

We were also confusing with this phenomenon before. Since the gas permeability of ionomers was tested in a film or membrane form, so the values reported here cannot 100% represents the actual gas permeability of ionomer in the catalyst layer. We thought that the water permeability of ionomers was much more important than the fuel gas permeability.

Figure R2. Ohmic resistance of PFTP-13, PFBP-14, PBP, PTP AEMs at 80 °C in OH⁻ form.

Table R2. The open cell voltage (OCV) of different AEIs-based AEMFCs with Pt/C.

Sample	PBP	PFBP-14	PTP	PFTP-13
OCV before activation	0.93~1.0	0.93~1.0	0.93~1.0	0.93~1.0
OCV after activation	0.97~1.03	0.98~1.06	0.97~1.02	0.97~1.05

Third, the authors indicated that PFPN-*x* polymers did not perform well in spite of low phenyl adsorption. The authors speculated that low performance is due to the insufficient molecular weight. I do not know this is true or not because the authors did not prove much of property data of PFPN-*x* (conductivity, mechanical property, water uptake). Moreover, the authors did not provide any molecular weight effect on performance. I do not believe, PFPN-*x* polymers are ready to publish in this manuscript.

Response: Thank you for reviewer's comment. Actually, we have done a lot of works on the PFPN-*x* (PFPN-100 and PFPN-85) ionomers. Moreover, we actually provided a lot of data for PFPN-100 and PFPN-85 ionomers as far as we can, including NMR, IEC, water uptake, λ value, intrinsic viscosity $[\eta]$, molecular weight, and fuel cell data. Please find the detailed information in Supplementary Table 2 and Supplementary Table 5. We have tested a lot of AEMFC

performance based on PFPN-100 and PFPN-85 ionomers, repeating up to 18 times. We also investigated the intrinsic viscosity effect of PFBP-*x* on fuel cell performance (PPDs) in Figure 4c in the manuscript. Actually, we also have SEM, TEM, SAXS, AFM, GPC, BET data of these ionomers, while we thought these data are unnecessary for this paper since there are too many characteristics as reviewer pointed out.

At the beginning of our research, we paid a lot of attention to PFPN-*x*, and we thought these two ionomers (PFBP-85 and PFBP-100) should possess excellent fuel cell performance before. Unfortunately, we cannot obtain these PFPN-*x* ionomers with sufficient intrinsic viscosity. Therefore, when PFPN-100 or PFPN-85 or high FL ratio of PFBP-*x* ionomers were used for MEA preparation, the catalyst layer is severely peeled off and detached from AEMs, resulting in poor fuel cell performance. Please find the detailed intrinsic viscosity effect in Figure 4c. **The low molecular weight of ionomers cannot bond the catalysts and AEMs well, which causes the detachment of catalyst layer.** This is a main issue in PFPN-*x* ionomers. In this case, PFPN-100 and PFPN-85 with low molecular weight showed poor film-forming properties and we cannot measure ion conductivity and mechanical properties. We do believe if we can synthesize PFPN-*x* ionomers with reasonable molecular weight, we can obtain excellent fuel cell performance, which will be our future work. We provided a detailed explanation for PFPN-*x* in page 14 in manuscript, as follows:

“Unfortunately, PFPN-100 and PFPN-85 AEIs with low adsorption and high rigidity, that are supposed to possess preferable cell performance, actually exhibited poor PPDs in both PFTP-13 and FAA-3-20 AEMs (0.3-0.7 W cm⁻²) due to insufficient molecular weight or intrinsic viscosity. The main reason is that PFPN-*x* ionomers cannot well bond the catalyst and AEMs, resulting in severe catalyst detachment from MEA during ion-exchange process.”

⊕ Supplementary Table 2. IEC, WU, SR, hydration number (λ), OH⁻ conductivity, and η of AEPs

Samples	IEC (mmol g ⁻¹)			WU (%) ^a	SR (%) ^a	λ	N_{free}	N_{bound}	$\sigma(\text{OH}^-)$ (mS cm ⁻¹) ^a	[η] (dL g ⁻¹) ^b
	Theo (OH ⁻)	Theo (I ⁻)	NMR (I ⁻)							
PBP	3.52	2.54	2.54	350±20	107±5	55	31	24	58±2	5.23
PFPN-100	3.09	2.31	2.31	78±12	—	14	—	—	NA	0.28
PFBP-14	3.43	2.49	2.49	320±15	94±5	52	21	31	63±3	2.34
PFPN-85	2.86	2.25	2.25	46±12	—	9	—	—	NA	0.38

Supplementary Table 5. Summary of fuel cell performance in this work along with number of repetitions in this work and for typical AEMFCs^{1,11-20}.

AEIs	AEMs	PPD (W cm ⁻²)	Catalyst types		Specific power (W mg ⁻¹)	Repeat times	Ref
			Pt/C anode	Pt-Ru/C anode			
Fumion	FAA-3-20	0.3 to 0.5 (with BP)	0.33 mg cm ⁻²		0.9 to 1.51	>10	This work
	FAA-3-50	0.3 to 0.4 (with BP)	0.33 mg cm ⁻²		0.9 to 1.21	>5	This work
	PFTP-13	0.4 to 0.65 (with BP)	0.33 mg cm ⁻²		1.21 to 1.96	3	This work
	PTP	0.4 to 0.5 (with BP)	0.33 mg cm ⁻²		1.21 to 1.51	2	This work
PFPN-85	FAA-20	0.15 to 0.35 (with BP)	0.33 mg cm ⁻²		0.45 to 1.06	>5	This work
	PFTP-13	0.25 to 0.65 (with BP)	0.33 mg cm ⁻²		0.76 to 1.97	>5	This work
PFPN-100	FAA-20	0.20 to 0.40 (with BP)	0.33 mg cm ⁻²		0.61 to 1.21	3	This work
	PFTP-13	0.300 to 0.68 (with BP)	0.33 mg cm ⁻²		0.91 to 2.06	>5	This work

Fourth, Figure 4 shows all ionomer effect since the same membrane PFTP-13 AEMs are used. The conclusion is reasonable that the cell using PFBP-14 AEIs performed well. However, if I read the main characterization part, PFTP-13 has more desirable properties. The liquid water uptake of PFBP-14 is very high which may result in electrode flooding. Water permeability of PFBP-14 is high which is good for electrode but high water permeability is also required for AEM. The high water permeability PFBP-14 is probably due to high liquid water uptake (then again it is not a good property due to potential flooding issue). H₂ permeability of PFBP-14 is much lower than PFTP-13, which is a significant disadvantage as an ionomer. Hydroxide conductivity of PFBP-14 is also lower than PFTP-13. I agree that high mechanical properties are required more with AEM but PFTP-13 has better mechanical properties than PFBP-14. So it seemed to yield that most properties of PFBP-14 are not desirable compared to the PFTP-13 AEM, then suddenly the best performance was achieved with PFBP-14 in Figure 14. I do not find any electrochemical half-cell data of PFBP-14 so it is puzzling why PFBP-14 performs the best.

Response: Thank you for reviewer's excellent question. Comment 2 and Comment 4 are helpful for our research.

Actually, we also have wondered why the PFBP-14 ionomer always showed the highest AEMFC performance in all AEMs in our group, even though we have other AEIs with higher ion conductivity. Results in this study show that high performance of PFBP-14 is truly due to the high-water permeability and reasonable WU of PFBP-14, along with matching RH conditions.

As mentioned earlier, the sufficient WU and high-water permeability of AEIs combining with suitable anode RH is the most important factors for high performance AEMFCs operating in high current density. Let's go back to AEMFCs with low current density:

(1) As shown in Figure R3a below, in high RH (90/100% RH) and low current density region ($<2 \text{ W cm}^{-2}$), PFTP-13 ionomer (WU: $\sim 70\%$) actually showed similar PPDs to the PFBP-14 ionomer (WU: $\sim 320\%$) based on the commercial FAA-3-20 membrane. Both PFBP-14 and PFTP-13 ionomers showed slightly higher PPDs than PBP ionomer (WU: $\sim 400\%$). We believed this is due to the higher ion conductivity (see Fig. 3a and b) and low ionomer adsorption of PFBP-14 and PFTP-13 compared with PBP and PTP.

However, at high RH and moderate current density region ($\sim 3.5 \text{ A cm}^{-2}$) (Fig. 4a in the manuscript) or at low RHs (50%/80%) and high current density region ($>5 \text{ A cm}^{-2}$) (Figure R2b), the PFTP-13 ionomer with low WU showed limited PPDs due to the dry-out issue; PPD of PFTP-13 is lower than those of PFBP-14 and PBP ionomers with high WU. Therefore, high (or suitable) WU of ionomers is beneficial for low RH and high current density conditions, where water should be supplied from high water retaining ionomers. Based on this, the matching RH for high WU AEIs is a key point for high performance ionomer. We used 75/100% A/C RH as mentioned before. At 100/100% RH, we experienced the water flooding issue as expected by the reviewer.

Therefore, PFBP-14 ionomer possessed much high PPDs than the PFTP-14 in high current density region and moderate RH. This result has been well documented in our group and can be easily repeated. Related description is the same with Comment 2.

Figure R3. (a) AEMFC performance based on commercial FAA-3-20 membrane and different AEIs with A/C TKK Pt/C at 65 °C along with 90%/100% A/C RH and 1,000/1,000 mL min^{-1} A/C flow rate. (b) AEMFC performance based on a PFTP-13 membrane and different AEIs with A/C TKK Pt/C at 80 °C along with 50%/80% A/C RH and 1,000/1,000 mL min^{-1} A/C flow rate.

(2) Regarding the potential anode flooding in the PFBP-14 ionomer, we used the moderate anode RH (75%) to well tackle this issue. We have done a lot of work in RH optimization, while all the data have not been presented in this paper due to excessive data issue.

Figure R4 is our unpublished data to verify the flooding issue based on PFBP-14 ionomers and 75/100% A/C RH for reviewer's reference. We investigated the cell performance with low fuel flow rate based on different AEIs (Figure R3a and 3b). Among these AEIs, the WU ranking is: PDTP-50 ($\sim 600\%$) $>$ PBP $>$ PFBP-14 $>$ PFTP-13, while the water permeability ranking is: PFBP-14 $>$ PBP $>$ PDTP-50 $>$ PFTP-13. The PFBP-14 ionomer shows the highest PPDs among different AEIs at low flow rate due to its high-water permeability which contributes to improving the water

back diffusion from the anode to cathode. Figure R3c and 3d shows the feasibility of AEMFCs operating in super low hydrogen flow rate below 100 mL min^{-1} . (Note: the quick voltage drop in Figure R3a is the discovery of limiting current density in low hydrogen flow rate, and the relationship between hydrogen flow rate and limiting current density will be published soon.)

As for the half-cell data of these ionomers, we measured the effect of PFBP, PFTP, PBP, PTP ionomers on HOR performance based on Pt/C, as shown in Figure R5 (This data is presented in Supplementary Fig. 23b along with related description). Basically, four types of AEIs didn't show significant adsorption or poisoning effect on HOR performance. FLN-containing AEIs (such as PFBP-14) showed slightly higher HOR current densities of Pt than pristine PBP or PTP AEIs due to the lower ionomer adsorption. This result is well matching with Y. S. Kim *et al*'s discovery (*Energy Environ. Sci.*, 2018, 11, 3283-3291).

Figure R4. AEMFC performance based on a PFTP-13 membrane and A/C Hispec Pt/C catalysts at 80°C with 75/100% A/C RH: a) different A/C ionomers with $100/100 \text{ mL min}^{-1} \text{ H}_2\text{-O}_2$ flow rate without backpressure at 80°C , b) different A/C ionomers with $150/1000 \text{ mL min}^{-1} \text{ H}_2\text{-O}_2$ flow rate with 1.3/1.3 A/C backpressure at 80°C , c) PFBP-14 ionomers with different A/C flow rates, d) *in-situ* durability testing of AEMFCs based on PFBP-14 A/C ionomers and a PFTP membrane ($30\pm 3\mu\text{m}$) with Pt/C catalysts under 0.3 A cm^{-2} at 70°C with $75/200 \text{ mL min}^{-1} \text{ H}_2/\text{O}_2$ flow rate and 90%/100% A/C RH.

Figure R5. HOR voltammograms of Pt/C microelectrodes in contact with four different ionomers. The voltammograms were measured in a saturated H_2 environment in 0.1 M KOH at 25 °C with a scan rate of 5 mV s^{-1} and a rotation speed of 2500 rpm.

This manuscript can be published in Nature Communications without an in-depth study on the durability portion (the first item). However, the authors may need to address the other weakness (No.2 – 4) before submitting the revised manuscript.

Response: Thank you for your comments. We tried our best to deal with reviewers' comments and I hope we have dealt with reviewers' comments adequately.

Reviewer #1 (Remarks to the Author):

Overall, I believe that the authors addressed the reviewer comments well. They were able to improve discussion on water transport. They also added relevant references. One thing that I still think should be done is an experiment that improves the durability. It is hard to write a paper that claims that the polymer is good for cell operation (water transport) and then blame poor cell operation for the lack of durability. That seems to be logic break. Other than that one experiment, this paper is fully ok to me.

Reviewer #2 (Remarks to the Author):

The authors have taken care of reviewers' questions. I noticed that the durability discussion has been substantially improved. However, the revised manuscript is not of the quality of Nature Communications and needs further evaluation.

First, the authors may need to examine the writing style to improve the clarity. I will take a few examples found in the abstract. In the abstract, the authors stated "outstanding ion conductivity" It should be stated to xx mS/cm at xx C instead of "outstanding ion conductivity". Also, it is not clear what the authors meant by "water management"? Does this mean by "flooding", "dry-out", "water uptake", "water permeability of AEM", or everything combined? The authors claimed the high performance was obtained by optimum AEMs and AEIs. However, it is not clearly demonstrated what is the process of optimization to select the AEMs and AEIs. The authors stated in the text that "the detailed optimization will be presented in our future work" "The durability of the present cells is limited by improper and un-optimized operating conditions,..." So the reviewer is confusing whether the authors did the optimization for the AEMs and AEIs under non-optimized conditions. Probably rational design is the right wording for materials.

Second, I found there are some technical items that the authors further need to explain. Mostly minor issues but there are some major issues as well.

1. (minor) Fig. 1. The chemical structures and IECs of PBP and PTP should be in Fig. 1 as their properties are compared in the text.
2. (minor) Fig. 2. First, the plots are confusing because the red font was used for PFBP-14 in Fig. 2b, but the blue font was used for PFBP-14 in Fig. 2c, 2d, and 2e.
3. (minor) Fig. 2. The reviewer is unsure what the benefit is to show Nafion property in this section. Nafion is an acid membrane and has a very different chemical structure. Including Nafion data makes the manuscript complex and distracting.
4. (minor) Fig. 2d. The zero water permeability at RH feed of 0 misleads readers. The zero value may be due to the experimental set-up where there is no water to permeate. Not the meaning that the permeability is zero. Those Zero values at 0% RH may be removed.
5. (major) Fig. 2d. The data showed that PFBP has higher water permeability and relatively lower hydrogen permeability than PFTP. Then PFBP is better for use of AEM and PFTP is better for use

of AEI, from the water management viewpoint. AEMFCs need a highly water permeable membrane that can remove water from the anode to supply the cathode. However, the authors stated that PFBP is for AEI and PFTP is for AEM. The water sorption data in Fig. 2c indicate that the high water permeability of PFBP-14 originates from high water sorption that probably causes water flooding. Therefore high water permeability of PFBP-14 should not help much for electrode performance. It is difficult to find the AEMFC data of MEAs (PFBP AEM/PFTP AEI vs. PFTP AEM/PFBP AEI). So I do not find strong evidence why PFTP works well with AEM and PFBT works well with AEI. However, one clear thing is PFBP has not enough mechanical property for the use of AEM (which is a good reason why use PFBP as an ionomer). But in this case, I do not think that the water management argument is well supported by the experimental evidence.

6. (minor) Fig. 3a. The authors should mention why no hydroxide conductivity data for PFBP-14, PBP, and Fumion at 90 and 100 C.
7. (major) This section clearly shows that PFTP membranes are advantageous for AEM use due to their high mechanical properties. However, the next section "AEMFC performance" does not explain well why PFBP performs higher than PFTP. The authors explained that high water uptake of PFBP-14 ionomer can help the performance. The reviewer does not agree because of the following reasons.
 - a. Normally, high water uptake causes electrode flooding and decreases fuel cell performance.
 - b. The polarization curves do not indicate any flooding-related performance loss.
 - c. The polarization curves do not indicate that the low performance of PFTP fuel cells is due to the severe electrode dry-out. The authors do not provide impedance that supports the dry-out of the cell. In addition, if there is a dry-out, it can be resolved by adjusting humidification. Moreover, if there is a dry-out in the electrode, that should happen at high current density regions. But the polarization curves of PFTP (compared to PFBP) cell has much lower performance at low current density regions. Therefore, the authors' explanation of water transport and dry-out arguments cannot be the reason for the low performance of PFTP ionomer cells. Having the reviewer's argument above regarding the choice of AEM and AEI, the data do not support that water management makes the difference in the performance between PFTP and PFBP cells.
 - d. I think a possible reason is phenyl adsorption as the authors mentioned in Ref. 37 and 38. Typically biphenyl based ionomer showed 400 - 500 mW/cm² higher PPD under H₂/O₂ conditions at 80 C because of less phenyl adsorption of biphenyl unit on the surface of Pt or Pt-Ru at the anode. Unfortunately, HOR voltammograms shown in Figure S23 was not able to pick this up because the RDE experiments run with 0.1 M KOH.
8. (minor) It is not clear why PFPNs did not work well, although the authors' explanation is reasonable. Without much information on water uptake, conductivity, adhesion properties, etc., the explanation is still speculative. So I suggest removing the PFPN.
9. (minor) The fuel cell performance demonstrated in Fig. 4d is impressive, particularly with relatively low loading of Pt. The authors should provide more information on how they measure the catalyst loading with an error range. Do the authors expect higher performance with higher catalyst loading?
10. (minor) I do not like to bring up specific power in this manuscript. Specific power depends on Pt catalyst loading. The high specific power for the cell demonstrated here is partly because the cell used relatively low Pt loading. If the authors add 0.03 mgPt/cm² to the Co@C/C cathode, the cell may have three times higher SP. The bottom line is the specific power reported here could be easily beaten by others when they use non-PGM catalyst with a little bit of Pt catalyst (see Reviewer 1 comment 9).

RESPONSE TO REVIEWERS

Manuscript Number: NCOMMS-20-39303A

Title: Durable poly(fluorenyl aryl piperidinium) membranes and ionomers for anion exchange membrane fuel cells

Authors: Nanjun Chen[†], Ho Hyun Wang[†], Sun Pyo Kim, Hae Min Kim, Won Hee Lee, Chuan Hu, Joon Yong Bae, Eun Seob Sim, Yong-Chae Chung, Jue-Hyuk Jang, Sung Jong Yoo, Yongbing Zhuang, Young Moo Lee*

We appreciate Editor and Reviewers for their time and efforts to provide critical comments to improve the present manuscript. We tried our best to answer all the questions from Reviewers as can be found in the following responses written in blue. All the changes are made in the revised version marked in yellow. Marked in red below is the newly added sentences in the revised version.

Reviewer #1

Overall, I believe that the authors addressed the reviewer comments well. They were able to improve discussion on water transport. They also added relevant references. One thing that I still think should be done is an experiment that improves the durability. It is hard to write a paper that claims that the polymer is good for cell operation (water transport) and then blame poor cell operation for the lack of durability. That seems to be logic break. Other than that one experiment, this paper is fully ok to me.

Response: Thank you for your comments. Regarding the durability, we presented our up-to-date durability data using our PFBP ionomer and PFTP membrane as shown in Figure R1 below (new Fig. 5b).

In collaboration with our collaborators and co-authors at Fuel Cell Research Center in Korea Institute of Science and Technology (KIST), based on PFBP-14 AEIs and a PFTP-13 membrane, our AEMFCs can be operated under a 1 A cm⁻² current density for >1000 h at 60 °C.

We replaced our *in-situ* durability data in Fig. 5b in the revised manuscript. Previous *in-situ* durability of AEMFCs for ~ 100 h measured at 80 °C and degradation of membranes from the autopsy of the MEA are now moved to Supporting information as Figures S29 and S30. We rephrased the *in-situ* durability part with the present data, which is highlighted in yellow in the revised manuscript.

Fig. R1. *in-situ* durability of H₂-O₂ AEMFCs based on PFBP-14 AEIs and a PFTP-13 AEM testing at a 1 A cm⁻² current density at 60 °C with 400/400 mL min⁻¹ A/C flow rate and 55 °C /62 °C A/C humidified temperatures without back pressure. The picture of the CCM after 1000 h *in-situ* durability testing is presented.

Reviewer #2 (Remarks to the Author):

The authors have taken care of reviewers' questions. I noticed that the durability discussion has been substantially improved. However, the revised manuscript is not of the quality of Nature Communications and needs further evaluation.

First, the authors may need to examine the writing style to improve the clarity. I will take a few examples found in the abstract. In the abstract, the authors stated "outstanding ion conductivity" It should be stated to xx mS/cm at xx C instead of "outstanding ion conductivity". Also, it is not clear what the authors meant by "water management"? Does this mean by "flooding", "dry-out", "water uptake", "water permeability of AEM", or everything combined? The authors claimed the high performance was obtained by optimum AEMs and AEIs. However, it is not clearly demonstrated what is the process of optimization to select the AEMs and AEIs. The authors stated in the text that "the detailed optimization will be presented in our future work" "The durability of the present cells is limited by improper and unoptimized operating conditions" So the reviewer is confusing whether the authors did the optimization for the AEMs and AEIs under non-optimized conditions. Probably rational design is the right wording for materials.

Response: Thank you for your comments, we rephrased these descriptions in the Abstract and in the main body by eliminating exaggerated and ambiguous words and descriptions and tried to clarify according to reviewer's comments. Changes are highlighted in yellow in the revised manuscript. The cell durability description in the Abstract has been rephrased since we provide a new 1000 h durability data.

Second, I found there are some technical items that the authors further need to explain. Mostly minor issues but there are some major issues as well.

1. (minor) Fig. 1. The chemical structures and IECs of PBP and PTP should be in Fig. 1 as their properties are compared in the text.

Response: We added the IECs of PBP and PTP in Fig. 1.

2. (minor) Fig. 2. First, the plots are confusing because the red font was used for PFBP-14 in Fig. 2b, but the blue font was used for PFBP-14 in Fig. 2c, 2d, and 2e.

Response: Thank you for pointing this out. We changed the color of PFBP-14 and PFTP-13 in Fig. 2b to match all figures.

3. (minor) Fig. 2. The reviewer is unsure what the benefit is to show Nafion property in this section. Nafion is an acid membrane and has a very different chemical structure. Including Nafion data makes the manuscript complex and distracting.

Response: We removed the Nafion data in Fig. 2 and the related description in the manuscript, and remained the supporting information data on Nafion for reader's reference.

4. (minor) Fig. 2d. The zero-water permeability at RH feed of 0 misleads readers. The zero value may be due to the experimental set-up where there is no water to permeate. Not the meaning that the permeability is zero. Those Zero values at 0% RH may be removed.

Response: We removed 0% RH data in Fig. 2d according to reviewer's suggestion.

5. (major) Fig. 2d. The data showed that PFBP has higher water permeability and relatively lower hydrogen permeability than PFTP. Then PFBP is better for use of AEM and PFTP is better for use of AEI, from the water management viewpoint. AEMFCs need a highly water permeable membrane that can remove water from the anode to supply the cathode. However, the authors stated that PFBP is for AEI and PFTP is for AEM. The water sorption data in Fig. 2c indicate that the high-water permeability of PFBP-14 originates from high water sorption that probably causes water flooding. Therefore, high water permeability of PFBP-14 should not help much for electrode performance. It is difficult to find the AEMFC data of MEAs (PFBP AEM/PFTP AEI vs. PFTP AEM/PFBP AEI). Therefore, I do not find strong evidence why PFTP works well with AEM and PFBP works well with AEI. However, one clear thing is PFBP has not enough mechanical property for the use of AEM (which is a good reason why use PFBP as an ionomer). But in this case, I do not think that the water management argument is well supported by the experimental evidence.

Response: Thank you for reviewer's further concerns to clarify the role of PFTP-13 and PFBP-14.

(1) Why selecting PFTP-13 as an AEM instead of PFBP-14?

First of all, for AEM application, we think low swelling ratio, excellent dimensional stability in OH^- form, high ion conductivity are the most important characteristics. Although PFBP-14 can form a film, the PFBP-14 membrane with swelling ratio over 120% at 80°C is not suitable for AEMs due to the limited dimensional stability. Actually, we believed mechanical properties of PFBP-14 was fine (50 MPa tensile strength in PFBP— I^- form—is actually pretty high), but the dimensional stability of PFBP in OH^- form was a big problem for AEM application. We never considered using highly swollen polymers as AEMs since we thought the precondition of high-performance AEMs was low swelling (<40%). Therefore, PFTP membrane with excellent mechanical properties, ion conductivity, low swelling ratio is the best choice for AEM applications. For further clarification, we revised some sentences in the manuscript as follows:

“On these grounds, the PFTP-13 copolymer with low SR, excellent dimensional stability, and high ion conductivity was chosen as representative AEMs, and PFBP-14 with high ion conductivity, high water permeability (or water diffusivity), but limited dimensional stability was selected as representative AEIs in this work.”

(2) Importance of water permeability for both AEMs and AEIs.

We are sorry that our previous description in PFBP-14 part is confusing. We agree with the Reviewer’s opinion that AEMs require high water permeability. Note that the water permeability (or water diffusivity) of PFTP-13 membrane is not lower at all (~20,000 Barrer at 80 °C). However, for AEMs, one of the most important aspect is low swelling ratio and excellent dimensional stability, and of course the WU (which my influence the water permeability) should not be too high. “Water permeability” and “water diffusivity” were separately measured and calculated as reported in Figure 2 and Table S4 in Supporting information, respectively. Both PFBP-14 and PFTP-13 AEIs show higher water diffusivity compared to PBP and PTP, respectively, which was highlighted according to *Reviewer #1’s* suggestion in the first round of revision.

Although water permeability of AEMs is important, our point is “high water permeability (or water diffusivity) of AEIs” is also very important for AEMFCs. We thought high water permeability of AEIs contribute to rapidly discharging the generated water in the anode which improves the water back diffusion to the cathode or to maintaining the water content in the cathode at low RH conditions or at water scarce condition.

(3) Any anode flooding issues in PFBP-14 AEIs?

Regarding the water behavior of ionomers, the water permeability of AEIs is impacted by the WU. “*Reviewer #2* thought high WU AEIs are easy to get flooding and not good for AEMFCs. We agree in some sense, but we have to define “high WU” on “how high it is” (~300%, ~500%, or ~2000%)?”

One cannot generalize that the high WU AEIs are not good for AEMFC because of flooding as we have two different types of ionomers: 1) rigid main-chain-type ionomers and 2) side-chain type ionomers. We think that main-chain-type ionomers (such as the present PFAP ionomers) and side-chain-type ionomers (which is on-going right now) seem to be quite different in terms of WU and SR etc. As shown in Fig. R2 from our preliminary side-chain type PFAP ionomer case, s-PFBP ionomers showed higher WU and SR than main-chain type PFBP at similar IECs since the flexible side chain in s-PFBP decreases the π - π stacking between the rigid backbones. We found that the phenomenon in s-PFBP seems well match with reviewer’s opinion that “high WU” s-PFBP ionomers are easy to get flooding. **Typical example is that the state-of-the-art FLN-100 ionomer (Fig. R3) produced by Y. S. Kim’s group (*Energy Environ. Sci.*, 2018, 11, 3283-3291). At similar IEC, FLN-100 (IEC: 3.5 mmol g⁻¹) exhibited much higher WU>2000% and unmeasurable SR at room temperature, while PFBP-14 ionomer (IEC: 3.43 mmol g⁻¹) showed WU of ~300% and ~100% SR at room temperature.** We must say that this is great merits of our PFBP-14 ionomers with reasonable dimensional stability. The results presented here may not be similar to these side-chain type ionomers or flexible ionomers (BTMA-ETFE) **since these flexible ionomers actually possess overlarge WU and poor dimensional stability in high IEC, which causes severe flooding issue.** Again, FLN-55 in Y. S. Kim’s EES paper showed >190% WU and 60% SR at 2.5 mmol g⁻¹ IEC, while our PFTP-13 ionomer showed only ~45% WU and ~16% SR at 2.82 mmol g⁻¹ IEC at room temperature. **The WU of AEIs lower than 50% is actually too low for AEMFCs operated at low RH** (due to low water permeability). Note that Y. S. Kim’s EES paper

didn't use FLN-40 (~75% WU) or FLN-30 (~34% WU) as ionomers (see Fig. R3), and they used FLN-55 (~190% WU) or FLN-100 (~2000% WU) as AEIs.

Figure R2. The chemical structure of side-chain-type PFBP ionomers.

Table 1 Electrochemical properties of ionomers used for this study

Ionomer	IEC (meq. g ⁻¹)		WU ^a (wt%)	Swelling (%)	σ (mS cm ⁻¹)		E_a (kJ mol ⁻¹)
	Theo.	NMR			30 °C	80 °C	
FLN-30	1.5	1.6	30 ± 4	7	36 ± 5	105 ± 5	19.0
FLN-40	2.0	2.0	75 ± 7	13	55 ± 5	119 ± 5	13.7
FLN-55	2.5	2.5	180 ± 13	60	110 ± 5	120 ± 7	1.5
FLN-100	3.5	3.5	>2000 ^b	NA ^c	NA	NA	NA
BPN ^d	2.6	2.6	145	40	62	127	12.7

^a Measured in OH⁻ form at room temperature. ^b Gel formation. ^c NA: not available. ^d Taken from ref. 32.

Fig. R3. The structure of FLN-100 and related physical properties in Y. S. Kim's publication [*Energy Environ. Sci.*, **2018**, 11, 3283-3291].

In summary, the cell behavior of flexible side-chain type ionomers could be different from these rigid main-chain type PFAP ionomers. We do believe our discovery is very useful for poly(aryl piperidinium)s family, such as Y. S. Yan [*Nature Energy*, **2019**, 4(5), 392-398] and L. Zhuang's AEMs and ionomers [*Angew. Chem. Int. Ed.* 2019, 58, 1442–1446]. We thought that the flooding is not an issue in our rigid PFBP-14 ionomers (excellent dimensional stability and water permeability). Moreover, Y. S. Yan *et al.* used PBP ionomers (IEC: 3.52 mmol g⁻¹) for their AEMFCs and no flooding issue was reported in *Nature Energy* paper. Actually, the WU of rigid PFBP-14 (~300%) is not that high compared to FLN-100 at the same IEC (>2000%). Since there is no water permeability data for the state-of-the-art FLN-55, FLN-100, BTMA-ETFE, or GT-x ionomers, it is impossible to fairly compare their performance. In our discovery, high water permeable (or water diffusivity) PFBP-14 shows the highest PPDs which is matching well with Reviewer #1's opinion.

Therefore, we believe AEIs with moderate WU and high-water permeability (or water diffusivity) is truly beneficial for ionomers and AEMs, while PFTP-13 ionomer with low WU (~45%) is not good for AEMFCs operated at low RH since it shows low water permeability at low RH.

We provide a new cell durability data with 1000 h in Fig. R1, which indicates that PFBP-14 AEIs-based AEMFCs can be operated at high current density (1 A cm⁻²) over 1000 h. We believe this data well demonstrates the great merits of our PFAP copolymers.

6. (minor) Fig. 3a. The authors should mention why no hydroxide conductivity data for PFBP-14, PBP, and Fumion at 90 and 100 C.

Response: The reason why we have not measured the conductivity of PFBP and PBP membranes is due to overlarge SR and poor dimensional stability at that temperatures. Having said that, the ion conductivity of most AEMs was measured below 80 °C. The reason why we measured at 90 °C or even 98 °C for PFTP-13 is to compare our data with Y. S. Yan's PAP-TP-*x* membrane in *Nature Energy* paper.

For FAA membrane or Fumion ionomer, we thought it is meaningless to measure their conductivity at high temperature, which is so unstable under alkaline conditions, and their AEMFCs are difficult to be operated in high temperature (usually, ~ 60°C). We mentioned these in the revised manuscript in the caption of Figure 3, and highlighted in yellow.

7. (major) This section clearly shows that PFTP membranes are advantageous for AEM use due to their high mechanical properties. However, the next section "AEMFC performance" does not explain well why PFBP performs higher than PFTP. The authors explained that high water uptake of PFBP-14 ionomer can help the performance. The reviewer does not agree because of the following reasons.

- a. Normally, high water uptake causes electrode flooding and decreases fuel cell performance.
- b. The polarization curves do not indicate any flooding-related performance loss.
- c. The polarization curves do not indicate that the low performance of PFTP fuel cells is due to the severe electrode dry-out. The authors do not provide impedance that supports the dry-out of the cell. In addition, if there is a dry-out, it can be resolved by adjusting humidification. Moreover, if there is a dry-out in the electrode, that should happen at high current density regions. But the polarization curves of PFTP (compared to PFBP) cell has much lower performance at low current density regions. Therefore, the authors' explanation of water transport and dry-out arguments cannot be the reason for the low performance of PFTP ionomer cells. Having the reviewer's argument above regarding the choice of AEM and AEI, the data do not support that water management makes the difference in the performance between PFTP and PFBP cells.
- d. I think a possible reason is phenyl adsorption as the authors mentioned in Ref. 37 and 38. Typically, biphenyl-based ionomer showed 400 - 500 mW cm⁻² higher PPD under H₂/O₂ conditions at 80 °C because of less phenyl adsorption of biphenyl unit on the surface of Pt or Pt-Ru at the anode. Unfortunately, HOR voltammograms shown in Figure S23 was not able to pick this up because the RDE experiments run with 0.1 M KOH.

Response: Thank you for reviewer's constructive suggestions. This comment is basically the same as *Comment 5* that "why PFBP ionomers showed higher performance than PFTP ionomers". We are sorry that our description of "*high WU of AEIs are good for AEMFCs*" is not clear. We have well replied the reason in *Comment 5*. We changed our description to "*high P_{water} of PFBP-14 AEIs with moderate WU contributed to improving the water back diffusion at high current density or to maintaining the water content at low RH conditions*" in page 14 in the revised manuscript.

The HFR or ASR of AEMFCs is our current drawback, and the dry-out explanation was not clear. Therefore, we re-phrased our description to "*On the contrary, PFTP-13 ionomers with low WU and high conductivity actually showed limited AEMFC performance at low RH due to relatively low P_{water}*" in page 14 in the revised manuscript.

For HOR measurement, we followed Y. S. Kim *et al*'s method [*Energy Environ. Sci.*, **2018**, 11, 3283-3291]. Actually, from HOR (Fig. R5), PFBP-14 actually showed higher performance than PFTP-13 and PBP, which is basically matching with Kim *et al*'s results. They are pioneer in this area, and we totally agree with the phenyl adsorption effect of aryl ether free polyelectrolytes and

appreciated for their discovery. We re-phrased the discussion in ionomer research and highlighted the phenyl adsorption of PFBP-14 in the revised manuscript in page 14, as follows:

“On the other hand, at high RH and moderate current density ($\sim 3.5 \text{ A cm}^{-2}$) (Fig. 4a) or at low RHs (50%/80%) and high current density ($> 5 \text{ A cm}^{-2}$) (Fig. 4b), PFBP-14 (1.42 W cm^{-2}) and PBP (1.19 W cm^{-2}) exhibited higher PPDs than PFTP-13 (1.06 W cm^{-2}) and PTP (0.98 W cm^{-2}) due to lower phenyl adsorption and higher P_{water} (or high-water diffusivity). According to Matanovic et al’s discovery^{37,38}, poly(biphenyl)-based ionomers showed higher PPDs than poly(terphenyl)-based ionomers due to lower phenyl adsorption, which is basically matching with our results. Moreover, high P_{water} of PFBP-14 AEIs with moderate WU contribute to rapidly discharging the generated water in the anode which improved the water back diffusion or to maintaining the water content at low RH conditions. Actually, PFBP-14 ionomers ($\sim 300\%$ WU at room temperature) showed much lower WU compared to the state-of-the-art polyfluorene ionomers³⁴ (side-chain-type FLN-100, $\text{WU} > 2000\%$ at room temperature) at similar IECs ($\sim 3.45 \text{ mmol g}^{-1}$). Therefore, PFBP-14 ionomers with moderate WU didn’t raise the anode flooding issues in the present AEMFCs. On the contrary, PFTP-13 ionomers with low WU and high conductivity actually showed limited AEMFC performance at low RH due to relatively low P_{water} . Meanwhile, AEMs also require reasonable water permeability. Note that PFTP-13 AEMs display higher PPDs than the commercial FAA series14 and PTP AEMs due to higher ion conductivity, mechanical properties, and P_{water} (or water diffusivity) (Supplementary Figs. 24).”

8. (minor) It is not clear why PFPNs did not work well, although the authors' explanation is reasonable. Without much information on water uptake, conductivity, adhesion properties, etc., the explanation is still speculative. So, I suggest removing the PFPN.

Response: Thank you for your suggestion. For PFPN-x ionomers, we declared that limited cell performance of the PFPN-x ionomers was due to the insufficient viscosity (or molecular weight), so that they cannot tightly hold the catalyst particles during ion exchange process, resulting in catalyst peeling off, according to our experimental observation. We do believe that PFPN-x ionomers with sufficient molecular weight will possess good performance.

Also, we have water uptake data and DFT calculation in phenyl adsorption reported in Supporting information. We noticed that low molecular weight of PFPN-*x* is an issue. We need this part to explain “how low viscosity” of ionomer that could not tightly hold on catalysts in these PAPs since PAP polymers from *Jannasch*'s groups showed very low intrinsic viscosity (in ref 24. $\sim 0.2 \text{ dL g}^{-1}$ to $\sim 0.47 \text{ dL g}^{-1}$). Although they reported many interesting PAP polymers, but their fuel cell performance was not that high, attributed partly due to the low molecular weight or viscosity of the synthesized PAP ionomers. It is important to have reasonable molecular weight or viscosity of PAPs if PAP can work as AEIs. Moreover, we intend to increase the molecular weight or viscosity of PFPN-*x* with acceptable molecular weight in the future. Even though, the results are not nice in these PFPN ionomers, we would like to remain this PFTP-*x* in the revised manuscript. We added the description of PFTP-*x* as follows in the revised manuscript in page 14.

“Unfortunately, PFPN-100 and PFPN-85 AEIs with low phenyl adsorption and high rigidity (Supplementary Fig. 13 and Supplementary Fig. 14) possess insufficient intrinsic viscosity (or molecular weight), which cannot tightly hold on catalyst particles, resulting in severe catalyst detachment from catalyst-coated membrane (CCM) during ion-exchange process. Therefore, AEMFC performance of PFPN-*x* ionomers ($0.3\text{-}0.7 \text{ W cm}^{-2}$) is limited. Figure 4c shows that the PPDs of AEMFCs are significantly impacted by the $[\eta]$ of PFBP-*x* AEIs, and the detachment issue has been found when the $[\eta]$ is lower than 1 dL g^{-1} along with PPD decrease. Development of PFPN-*x* and PFBP-*x* ionomers with reasonable molecular weight will be our future work. These discoveries provide a clear information for the PAP family to rationally design AEIs with sufficient molecular weight”

9. (minor) The fuel cell performance demonstrated in Fig. 4d is impressive, particularly with relatively low loading of Pt. The authors should provide more information on how they measure the catalyst loading with an error range. Do the authors expect higher performance with higher catalyst loading?

Response: We calculated the catalyst loading by measuring the mass of CCM before and after along with many repeated experiments. For example, we finally obtained 12.5 mg cm^{-2} Hispec Pt/C, 12.5 mg Pt-Ru/C , 11.1 mg TKK Pt/C for $0.4 \pm 0.02 \text{ mg cm}^{-2}$ loading in 5 cm^2 CCM, which is similar with Kim *et al*'s results (*Journal of Materials Chemistry A*, 2020, 8, 14135-14144. 20 mg Pt-Ru/C for $0.75 \text{ mg}_{\text{Pt-Ru}} \text{ cm}^{-2}$ loading). When using CCM method, CCM will have a little bit of swelling after ion exchange process, and the catalyst area will be a little bit larger than 5 cm^2 . Therefore, we recorded the area swelling of CCM, and calculated the actual loading of MEA (5 cm^2) based on the area swelling.

We never tried high loading of 0.75 mg cm^{-2} , and our intention was to decrease the catalyst loading. When we slightly increased the Pt-Ru/C loading from 0.4 mg cm^{-1} to 0.5 mg cm^{-1} , the power density slightly increased. We thought high but suitable catalyst loading will be helpful for AEMFCs since many current research groups used high loading.

10. (minor) I do not like to bring up specific power in this manuscript. Specific power depends on Pt catalyst loading. The high specific power for the cell demonstrated here is partly because the cell used relatively low Pt loading. If the authors add 0.03 mgPt/cm^2 to the Co@C/C cathode, the cell may have three times higher SP. The bottom line is the specific power reported here could be easily beaten by others when they use non-PGM catalyst with a little bit of Pt catalyst (see Reviewer 1 comment 9).

Response: Thank you for reviewer's suggestion. We agreed with *Reviewer #2*'s opinion. However, *Reviewer #1* suggested us to mention the specific power. Therefore, we bring down some

information in specific power and highlight the specific power based on the Pt loading as follows in page 16 of the revised manuscript:

“However, GT-*x* series and BTMA-HDPE-based AEMFCs actually employed a high Pt-Ru/C loading of 0.7 mg cm^{-2} , resulting in the limited specific power³³ between $4 \sim 5 \text{ W mg}^{-1}$. The present AEMFCs exhibited high PGM-based specific power (5.57 to 6.52 W mg^{-1}), which is even comparable to PGM-free AEMFCs (6 W mg^{-1} to 8 W mg^{-1})^{44,45}.”

We try our best to improve our manuscript and we hope our response can well answer reviewer's comments.

Reviewer #1 (Remarks to the Author):

As I stated in my previous remarks, the authors should present IMPROVED durability - not just longer duration. This is a real sticking point for me. I really like every other aspect of the paper, but losing 50% of the operating voltage over 1000 hours is very poor. It is also the same issue with the operating points, etc. As I said in my previous review, there is a disconnect here in logic where the operating conditions are likely causing the low performance, and "it is hard to write a paper that claims that the polymer is good for cell operation (water transport) and then blame poor cell operation for the lack of durability."

In my view, the authors need to take a new dataset that really shows good durability. Even if it is over only a couple hundred hours (1000 hours is more than a month and a lot can go wrong there). One thing that has been done in the past with other groups is to sacrifice some performance to achieve that durability - where the same operating conditions to get high power are not necessarily the same ones to get long life. A recent example of that is this paper: <https://onlinelibrary.wiley.com/doi/abs/10.1002/aenm.202001986>. Some people - myself included - criticize this group for doing that, but with a good discussion around it, such treatment can be justified.

Reviewer #2 (Remarks to the Author):

The authors took care of the reviewers' comments well. I do not 100% agree that the authors' answers regarding water transport without having data of PFBP AEM and PFTP ionomer. But I do agree that mechanical properties are the most critical property for AEM. As the authors mentioned the importance of the mechanical properties, I am fine with the revised version of the manuscript. Other answers are satisfactory.

I have two additional comments for the revised text.

1. Page 16. The authors mentioned the specific power of PGM-free AEMFC is 6 to 8 W/mg. If the MEA is PGM-free AEMFC then the specific power should be infinity because specific power is calculated to power per PGM catalyst loading. Please clarify how the references calculated the specific power. Do they use PGM-free cathode and calculate the specific power per anode and cathode?

2. Page 19. The authors indicated that high current density (over 4 A cm⁻²) represents harsh conditions. This assumption came from Dekel's paper that high current density conditions can dehydrate the cathode and accelerate degradation. There are a couple of arguments. First, Dekel did not show actual cell degradation. Most of his argument came from small molecule study and calculation data. Second, his low RH condition meant a lambda value of 4 or less. I do not think such a condition was applied in the cells. Third, the authors evaluated the durability at 0.2 A/cm² (Figure 30), not over 4 A/cm². Based on our experience, the 0.2 A/cm² condition is more detrimental because the cell voltage is over 0.6 V (cathode potential is even higher). In this condition, electrooxidation can occur. The 1,000 h durability was obtained mostly with lower potential so the degradation of the membrane may be less. Fourth, if degradation occurs with short expose at 4 A/cm² conditions, all of the authors' good performing MEAs in Figure 4a, b, d should also show degradation after I-V curves. If the degradation indeed occurs, then I would argue what is a good

thing with 2.3 W/cm² peak power where the degradation of the MEA is severe.

I am also wondering whether the authors obtained the I-V curves for the 1,000 h life test cell. Why the authors did the I-V curve measurement only for the 120 h test cell before the life test?

My suggestion to clarify this is as follows:

If the authors obtained the I-V curves for the 1000 h life test cell, it is much clear and can state "durability testing under 0.2 A cm⁻² at 80 C for 120 h showed 12.5 degradation, possibly due to electrochemical oxidation under higher cell voltage"

If the authors did not obtain the I-V curves for the 1000 h cell, then just stated the observed degradation with 120 h without saying that high current density causes this.

If the authors want to state that is due to high current density, then they need to get more data at a constant current density of 4 A/cm² to prove it.

Reviewer #1's comment (Second round)

Overall, I believe that the authors addressed the reviewer comments well. They were able to improve discussion on water transport. They also added relevant references. One thing that I still think should be done is an experiment that improves the durability. It is hard to write a paper that claims that the polymer is good for cell operation (water transport) and then blame poor cell operation for the lack of durability. That seems to be logic break. Other than that one experiment, this paper is fully ok to me.

Reviewer #1's comment (Third round):

As I stated in my previous remarks, the authors should present **IMPROVED** durability - not just longer duration. This is a real sticking point for me. **I really like every other aspect of the paper**, but losing 50% of the operating voltage over 1000 hours is very poor. It is also the same issue with the operating points, etc. As I said in my previous review, there is a disconnect here in logic where the operating conditions are likely causing the low performance, and "it is hard to write a paper that claims that the polymer is good for cell operation (water transport) and then blame poor cell operation for the lack of durability."

In my view, the authors need to take a new dataset that really shows good durability. **Even if it is over only a couple hundred hours** (1000 hours is more than a month and a lot can go wrong there). One thing that has been done in the past with other groups is to sacrifice some performance to achieve that durability-where the same operating conditions to get high power are not necessarily the same ones to get long life. A recent example of that is this paper: <https://onlinelibrary.wiley.com/doi/abs/10.1002/aenm.202001986>. Some people-myself included-criticize this group for doing that, but with a good discussion around it, such treatment can be justified.

Response: Thank you for reviewer's comments. We misunderstood the reviewer's intention about "improved durability". We understood that the good durability meant longer duration, as there was a report for 1000 h data reported by Leonard *et al.* (*J Material Chem. A*, 2020, **8**, 14135-14144) that showed about roughly 40% voltage loss during that period.

We replace the former durability data with new dataset with improved durability performance up to 200 hours in Figure R1 (Fig. 5b in the revised manuscript). The voltage loss was about 3.68% which was much stable compared to original one that we presented in our original submission and first revision. We have added and compared the discussion related with the durability of other AEMFCs reported recently in page 19 of the revised manuscript.

To eliminate the confusion, we deleted the discussion on degradation from the main text and supporting information.

Figure R1. *In-situ* durability of H₂-O₂ AEMFCs based on PFBP-14 AEs and PFTP-13 AEMs testing at a 0.2 A cm⁻² current density at 70 °C with a 100/100 mL min⁻¹ A/C flow rate and 90/100% A/C RH.

Reviewer #2's comment (Third round):

The authors took care of the reviewers' comments well. I do not 100% agree that the authors' answers regarding water transport without having data of PFBP AEM and PFTP ionomer. But I do agree that mechanical properties are the most critical property for AEM. As the authors mentioned the importance of the mechanical properties, I am fine with the revised version of the manuscript. Other answers are satisfactory.

Response: Thank you for your positive comments.

I have two additional comments for the revised text.

1. Page 16. The authors mentioned the specific power of PGM-free AEMFC is 6 to 8 W/mg. If the MEA is PGM-free AEMFC then the specific power should be infinity because specific power is calculated to power per PGM catalyst loading. Please clarify how the references calculated the specific power. Do they use PGM-free cathode and calculate the specific power per anode and cathode?

Response: Thank you for reviewer's comments. As Reviewer 2 indicated in Revision 2 and did not like to bring up the specific power in this manuscript, we would like to remove specific power portion from this manuscript to eliminate any confusion. We removed the related description from the manuscript and supporting information.

2. Page 19. The authors indicated that high current density (over 4 A cm⁻²) represents harsh conditions. This assumption came from Dekel's paper that high current density conditions can dehydrate the cathode and accelerate degradation. There are a couple of arguments. First, Dekel did not show actual cell degradation. Most of his argument came from small molecule study and

calculation data. Second, his low RH condition meant a lambda value of 4 or less. I do not think such a condition was applied in the cells. Third, the authors evaluated the durability at 0.2 A/cm² (Figure 30), not over 4 A/cm². Based on our experience, the 0.2 A/cm² condition is more detrimental because the cell voltage is over 0.6 V (cathode potential is even higher). In this condition, electrooxidation can occur. The 1,000 h durability was obtained mostly with lower potential so the degradation of the membrane may be less. Fourth, if degradation occurs with short expose at 4 A/cm² conditions, all of the authors' good performing MEAs in Figure 4a, b, d should also show degradation after I-V curves. If the degradation indeed occurs, then I would argue what is a good thing with 2.3 W/cm² peak power where the degradation of the MEA is severe.

Response: Thank you for reviewer's suggestion. The followings are the responses to Reviewer 2's comments.

(1) We are sorry to make confusion related with 4 A cm⁻² discussion. To eliminate the confusion, we deleted the description of "4 A cm⁻² current density is a harsh condition", from the revised manuscript,

(2) We agreed with Reviewer's opinion that lamda < 4 could not be applied for AEMFCs.

(3) Thank you for your constructive comments that 0.2 A cm⁻² current density is maybe more detrimental due to high voltage.

(4) The stability behavior of AEM/AEI under high current density should be thoroughly checked later as these are not the key points of this manuscript. Therefore, we deleted the related descriptions from the manuscript.

3. I am also wondering whether the authors obtained the I-V curves for the 1,000 h life test cell. Why the authors did the I-V curve measurement only for the 120 h test cell before the life test?

My suggestion to clarify this is as follows:

If the authors obtained the I-V curves for the 1000 h life test cell, it is much clear and can state "durability testing under 0.2 A cm⁻² at 80 C for 120 h showed 12.5 degradation, possibly due to electrochemical oxidation under higher cell voltage"

If the authors did not obtain the I-V curves for the 1000 h cell, then just stated the observed degradation with 120 h without saying that high current density causes this.

If the authors want to state that is due to high current density, then they need to get more data at a constant current density of 4 A/cm² to prove it.

Response: 1000 h durability data we presented in the second round revision was collected separately by our collaborators at KIST. To eliminate the confusion, we removed the 1000 h durability data and provided a new and improved durability data as presented in Figure 5b.

We appreciate the Editor and Reviewers for their time and efforts to improve the present manuscript. We hope we have dealt with Reviewers' comments adequately.

Reviewer #1 (Remarks to the Author):

I am in agreement with the changes here. I am ok with this being published without any further revision.

Reviewer #2 (Remarks to the Author):

I do not have major concerns. However, it seems that English can be improved. Also I suggest a few things for better reading.

Page 8 line 151. The authors stated that "the rate of water generation in the anode is two times faster..." Is that the rate? Or the amount of water? The reviewer believes the "amount", not "rate".

Fig. 1. I suggest changing the acronym of PBP to PFBP-0 and PTP to PFTP-0. Too many acronyms make it confusing.

Fig2 caption. Please remove the wording "commercial Nafion membranes were used for comparison"

Page 11 subtitle "Physical properties" I suggest "Electrochemical and physical properties" or "Conductivity and mechanical properties"

Page 11 line 212. remove "truly" from PFAP backbone is truly beneficial..

Supplementary Table 5. Please remove Tg column (not informative because no data for other membranes) but include testing temperature and RH conditions which impact the mechanical properties substantially.

Page 14 line 274 remove "actually"

Page 16 line 321 What is the "power cost"? Consider rephrasing the sentence to "While the PFAP-based cells exhibited high performance with a low PGM loading of"

Figure 4 figure caption.

Suggest the figure caption title to "Fuel cell performance"

4c: x-axes: correct to FLN ratio.

Figure 4a and 4b are polarization curves and power density. Also the introducing statement "The power density of AEMFCs with..." looks to apply to all figures but only belongs to a and b.

Figure 4e Figure caption says "same testing conditions as d." However, in the figures, the backpressure between d and e are different. Also, Fig 4e is oxygen condition and Fig.4e is air condition.

Scheme 1. I am not sure why the authors called this a scheme. This is a figure.

"pick circle"?

What are the brown star and red circle represent?

Reviewer #1's comment

I am in agreement with the changes here. I am ok with this being published without any further revision.

Response: Thank you for your time and constructive comments that improved the manuscript.

Reviewer #2's comment

I do not have major concerns. However, it seems that English can be improved. Also I suggest a few things for better reading.

Response: Thank you for your positive comments. We polished the language according to Reviewer's suggestions.

Page 8 line 151. The authors stated that "the rate of water generation in the anode is two times faster..." Is that the rate? Or the amount of water? The reviewer believes the "amount", not "rate".

Response: We changed the "rate" to "amount" in page 7 in the revised manuscript.

Fig. 1. I suggest changing the acronym of PBP to PFBP-0 and PTP to PFTP-0. Too many acronyms make it confusing.

Response: We changed the acronym of PBP to PFBP-0 and PTP to PFTP-0 according to reviewer's suggestion in the revised manuscript and Supporting Information.

Fig 2 caption. Please remove the wording "commercial Nafion membranes were used for comparison"

Response: We deleted this sentence in Fig. 2 caption in the revised manuscript.

Page 11 subtitle "Physical properties" I suggest "Electrochemical and physical properties" or "Conductivity and mechanical properties"

Response: We changed the subtitle of "Physical properties" to "Electrochemical and physical properties". in page 11.

Page 11 line 212. remove "truly" from PFAP backbone is truly beneficial.

Response: We removed this word in this sentence in page 11.

Supplementary Table 5. Please remove Tg column (not informative because no data for other membranes) but include testing temperature and RH conditions which impact the mechanical properties substantially.

Response: We removed Tg in Supplementary Table 5 according to reviewer's comments.

Page 14 line 274 remove "actually"

Response: We removed this word in page 14 in revised manuscript.

Page 16 line 321 What is the "power cost"? Consider rephrasing the sentence to "While the PFAP-based cells exhibited high performance with a low PGM loading of"

Response: We refreshed the sentence to "while the PFAP-based cells exhibited high performance with a low PGM loading of $0.33 \text{ mg cm}^{-2} \sim 0.42 \text{ mg cm}^{-2}$ according to Reviewer's suggestion.

Figure 4 figure caption.

Suggest the figure caption title to "Fuel cell performance"

4c: x-axes: correct to FLN ratio.

Figure 4a and 4b are polarization curves and power density. Also the introducing statement "The power density of AEMFCs with..." looks to apply to all figures but only belongs to a and b.

Figure 4e Figure caption says "same testing conditions as d." However, in the figures, the backpressure between d and e are different. Also, Fig 4e is oxygen condition and Fig.4e is air condition.

Response:

(1) We changed the figure caption title to "Fuel cell performance" according to Reviewer's suggestion.

(2) We corrected the typing mistake to FLN ratio in Figure 4c in x-axis.

(3) We changed the expression of Figure 4 caption by separating each caption descriptions to avoid confusion according to Reviewer's comments as follows:

Fig. 4 | Fuel cell performance. **a**, the power density of AEMFCs with different AEIs and PFTP-13 AEMs ($25 \pm 3 \mu\text{m}$) based on A/C TKK Pt/C catalysts (0.33 mg cm^{-2}): 1,000/1,000 mL min^{-1} $\text{H}_2\text{-O}_2$ flow rate, $65 \text{ }^\circ\text{C}$, 85%/100% A/C RH, and 0.5/0.5 bar back pressure. **b**, the power density of AEMFCs with different AEIs and PFTP-13 AEMs ($25 \pm 3 \mu\text{m}$) based on A/C TKK Pt/C catalysts (0.33 mg cm^{-2}): $80 \text{ }^\circ\text{C}$, 50%/80% A/C RH, and 1.3/1.3 bar back pressure. **c**, the relationship between the PPDs and the intrinsic viscosity of PFBP-*x* AEIs: PFTP-13 AEMs ($25 \pm 3 \mu\text{m}$), $\text{H}_2\text{-O}_2$, A/C Hispec Pt/C catalysts (0.33 mg cm^{-2}), $80 \text{ }^\circ\text{C}$, A/C flow rate of 1,000/1,000 mL min^{-1} , 75%/100% A/C RH. Hollow circle symbols are PPDs without back pressure, while filled circle symbols are PPDs with 1.3/1.3 bar back pressure. **d**, the power density of AEMFCs based on Pt-Ru/C anode with back pressure. PFBP-14 AEIs and PFTP-13 AEMs ($20 \pm 3 \mu\text{m}$), $80 \text{ }^\circ\text{C}$, 75%/100% A/C RH, 1,000/1,000 mL min^{-1} $\text{H}_2\text{-O}_2$ flow rate, different back pressures, Pt-Ru/C anode (0.42 mg cm^{-2}), Hispec Pt/C cathode (0.33 mg cm^{-2}). A/C Hispec Pt/C (0.33 mg cm^{-2}) for comparison. **e**, the power density of AEMFCs in $\text{H}_2\text{-air}$ (CO_2 free) with different anode catalysts: PFBP-14 AEIs and PFTP-13 AEMs ($20 \pm 3 \mu\text{m}$), $80 \text{ }^\circ\text{C}$, 75%/100% A/C RH, 1,000/2,000 mL min^{-1} flow rate, 1.3/1.3 bar A/C back pressure, 0.33 mg cm^{-2} A/C catalyst loading.

Scheme 1. I am not sure why the authors called this a scheme. This is a figure.

"pick circle"?

What are the brown star and red circle represent?

Response:

(1) We changed the title of "Scheme 1" to "Fig. 5" in the revised manuscript according to reviewer's comment.

(2) We are sorry for our typing mistake of "pick circle" which should be "pink circle". We collected it in Fig. 5 caption.

(3) Brown star and red circle represent the FAA Fumion and PFAP ionomers in this work, respectively, and the related descriptions were added to Fig. 5 caption in the revised manuscript.